# ATP synthase F$_1$ subunits recruited to centromeres by CENP-A are required for male meiosis

Caitríona M. Collins [1], Beatrice Malacrida[2], Colin Burke[1,3], Patrick A. Kiely[2] & Elaine M. Dunleavy[1]

The histone H3 variant CENP-A epigenetically defines the centromere and is critical for chromosome segregation. Here we report an interaction between CENP-A and subunits of the mitochondrial ATP synthase complex in the germline of male *Drosophila*. Furthermore, we report that knockdown of CENP-A, as well as subunits ATPsyn-α, -βlike (a testis-specific paralogue of ATPsyn-β) and -γ disrupts sister centromere cohesion in meiotic prophase I. We find that this disruption is likely independent of reduced ATP levels. We identify that ATPsyn-α and -βlike localise to meiotic centromeres and that this localisation is dependent on the presence of CENP-A. We show that ATPsyn-α directly interacts with the N-terminus of CENP-A in vitro and that truncation of its N terminus perturbs sister centromere cohesion in prophase I. We propose that the CENP-A N-terminus recruits ATPsyn-α and -βlike to centromeres to promote sister centromere cohesion in a nuclear function that is independent of oxidative phosphorylation.

[1] Centre for Chromosome Biology, Biomedical Sciences, National University of Ireland Galway, Galway, Ireland H91TK33. [2] Graduate Entry Medical School and Health Research Institute, University of Limerick, Limerick V94 T9PX, Ireland. [3] Present address: Queen's University, Belfast BT7 1NN Northern Ireland, UK. Correspondence and requests for materials should be addressed to E.M.D.(email: elaine.dunleavy@nuigalway.ie)

Meiosis is the specialised cell division cycle in which one round of DNA replication precedes two rounds of chromosome segregation that generate haploid gametes (eggs and sperm). Defects in meiosis lead to reduced fertility, sterility or aneuploidy in gametes or resulting zygotes[1]. Centromeres, defined epigenetically by incorporation of the histone H3 variant CENP-A[2] play a key role in coordinating meiotic chromosome segregation. Studies in plants suggest that CENP-A adopts meiosis-specific functions via its highly divergent N terminus[3–6]. To investigate functions of the CENP-A N terminus in meiosis in an animal, we used biochemical and genetic approaches in testis of *Drosophila* males. We uncover unexpected functional links between CENP-A, mitochondrial ATP synthase $F_1$ subunits and sister centromere cohesion in meiosis. We propose that the CENP-A N-terminus recruits ATPsyn-α and -βlike, a testis-specific paralogue of -β, to centromeres to promote sister centromere cohesion in a novel nuclear function that is independent of canonical roles in oxidative phosphorylation.

## Results

**CENP-A functions in meiotic sister centromere cohesion**. To investigate meiosis-specific requirements for CENP-A in *Drosophila*, we performed testis-specific knockdown of CENP-A using the UAS-GAL4 RNAi system. We expressed GAL4 under the control of the bag of marbles (bam) promoter knocking down CENP-A in the mitotic divisions immediately prior to meiotic prophase I (Fig. 1a). In *Drosophila* males, meiotic prophase I lacks the conventional features of synapsis between homologues and instead is divided into sub-stages S1–S6 based on nuclear and spindle morphologies[7]. At S5/6, the four *Drosophila* chromosomes are separated into three territories; the 2nd and 3rd autosomes each form a large territory and the X–Y chromosomes form a third territory with the 4th chromosome (Fig. 1a). We immuno-stained RNAi-depleted S5/6 spermatocytes for the centromere markers CENP-A and CENP-C, confirming an ~30% reduction in CENP-A level at centromeres (Supplementary Fig. 1A). At S5/6, an average of 6.5 centromere foci are normally visible; two each per 2nd/3rd autosomal territories, one or two per 4th chromosomal territory and one or two per X–Y territory[8,9] (Fig. 1a). In S6 nuclei depleted for CENP-A an unexpected increase in the number of centromere foci was observed compared to the control (Fig. 1b). Quantitation of centromere foci per nucleus revealed a significant increase (****$p$ < 0.0001) at late prophase I (S5/6, 7.7 compared to 6.5 in the control) and prometaphase I (7.1 compared to 5.7 in the control) (Fig. 1c). As homologues are normally unpaired at this stage (except the 4th chromosome), these results suggest that the maintenance of sister centromere cohesion at late prophase I is defective upon CENP-A reduction. This phenotype was enhanced with increased RNAi efficiency in crosses performed at 29 °C (Supplementary Fig. 1A, 1B, 1C); an almost 50% reduction in CENP-A at centromeres resulted in 9.1 foci per S5/6 nucleus compared to 6.5 in the control (****$p$ < 0.0001). We also assayed the number of centromere foci at early prophase I (S1/2a). At this stage, on average three CENP-A spots are normally visible as homologues are paired, sister chromatid cohesion is intact and centromeres cluster non-specifically akin to chromocenters[8] (Fig. 1a). Quantitation of centromere foci per nucleus revealed a significant increase (****$p$ < 0.0001) at early prophase (S1/2a, 4.3 foci compared to 3.07 in the control), suggesting defects either in sister centromere cohesion or homologue pairing. The number of centromere foci detected per nucleus at interphase did not differ from the control (3.7 compared to 3.6, $p$ = 0.366). It is possible that these interphase cells derived from prophase I cells with normal cohesion, or that these cells arise due to compensation by

additional factors that maintain cohesion at this time. Additionally, it is also possible that the CENP-A depletion was less efficient in these cells.

**ATP synthase $F_1$ subunits co-purify with CENP-A**. In parallel, we aimed to identify novel regulators of CENP-A function in germ cells using a biochemical approach. Based on findings in plants in which either the deletion of the CENP-A N terminus or its replacement by that of histone H3.3 resulted in sterility[4,5], we hypothesised that the CENP-A N terminus is functionally important in *Drosophila* meiosis. To identify proteins interacting with the CENP-A N terminus, we made soluble extracts from wild-type adult fly testes and performed a pull-down using a recombinantly expressed CENP-A N terminus with a GST tag (GST-Nterm-CENP-A) or GST only as bait (Fig. 1d). Surprisingly, subunits of the $F_1$ portion of the mitochondrial ATP synthase complex V co-purified with GST-Nterm-CENP-A (Supplementary Data 1). ATP synthase normally functions in oxidative phosphorylation catalysing the synthesis of ATP from ADP and inorganic phosphate[10]. Yet, literature searches revealed previous links between the complex and male fertility in *Drosophila*. First, *ATPsyn-α* mutants are male sterile[11]. Second, *ATPsyn-β* expression is normally repressed in the testis and its derepression impairs fertility[12]. Third, we noted a testis-specific paralogue of *ATPsyn-β*, *ATPsyn-βlike*, previously identified in male-sterile screens[11,13]. ATPsyn-βlike is 70% homologous to ATPsyn-β, but harbours unique N and C terminal extensions (Supplementary Fig. 1D) and recent phylogenetic analysis detected *ATPsyn-βlike* in insect subgroups Diptera and Lepidoptera[14]. We confirmed the testis-specific expression of *ATPsyn-βlike* by RT-PCR (Supplementary Fig. 1E) and western analysis (Supplementary Fig. 1F). Finally, we noted that ATP synthase $F_1$ subunits (*ATPsyn-α*, -β and -γ) are functionally linked to germ line stem cell differentiation in *Drosophila* females[15,16]. Moreover, this unexpected function was proposed to be independent of canonical functions in oxidative phosphorylation[16]. Based on these findings, we investigated further the link between *ATPsyn-α*, -β, -βlike and -γ in male fertility, as well as potential links to meiotic centromere function.

**ATP synthase $F_1$ subunits are required for male fertility**. We first performed testis-specific RNAi for ATPsyn-α, -β, -βlike and −γ at 25 °C (or at 29 °C to enhance RNAi efficiency) and confirmed knockdowns by quantitative PCR (qPCR) (Supplementary Fig. 1G) and immuno-fluorescent microscopy for cytoplasmic signals (Supplementary Fig. 1H). Fertility assays showed that males depleted for ATPsyn-α, -βlike and -γ were sterile, while males depleted for ATPsyn-β had an ~20% reduction in fertility compared to the wild-type control (Supplementary Fig. 2A), in line with previous findings[11–13]. To account for fertility defects, we analysed meiotic cell cycle progression in ATPsyn-α, -β, -βlike and -γ RNAi-depleted testes. We counted the number of cysts undergoing meiosis I or II or containing spermatids in adult testes (Fig. 1e). Testes depleted for ATPsyn-α, -β or -γ at 25 °C did not differ from controls in the number of cysts undergoing meiosis I ($p$ = 0.922, $p$ = 0.717 and $p$ = 0.304, respectively), or meiosis II ($p$ = 0.242, $p$ = 0.203 and $p$ = 0.778, respectively). In testes depleted for ATPsyn-α and -β, the number of spermatid cysts was not significantly reduced ($p$ = 0.147 and $p$ = 0.646), however in testes depleted for ATPsyn-γ the number of spermatid cysts was significantly reduced (****$p$ < 0.0001) compared to the control. Strikingly, in RNAi experiments performed at 25 °C or 29 °C, testes depleted for ATPsyn-βlike arrest before the first meiotic division (Supplementary Fig. 2B) and no mature sperm is produced. ATPsyn-α RNAi performed at 29 °C

resulted in a significant decrease in the number of cysts undergoing meiosis II (*$p < 0.05$) and those containing spermatids (****$p < 0.0001$), yet mature sperm is produced. While spermatocytes depleted for ATPsyn-βlike arrest at prometaphase I, spermatocytes depleted for ATPsyn-α at 29 °C accumulate at prometaphase I with perturbed nuclear morphologies yet

abnormal divisions proceed (Supplementary Fig. 2B). Bright-field imaging of adult testis depleted for ATPsyn-βlike showed an abnormal morphology and confirmed the absence of mature spermatozoa from seminal vesicles (Fig. 1f). In contrast, seminal vesicles of ATPsyn-α, -β and -γ RNAi-depleted samples were plump and contained mature sperm.

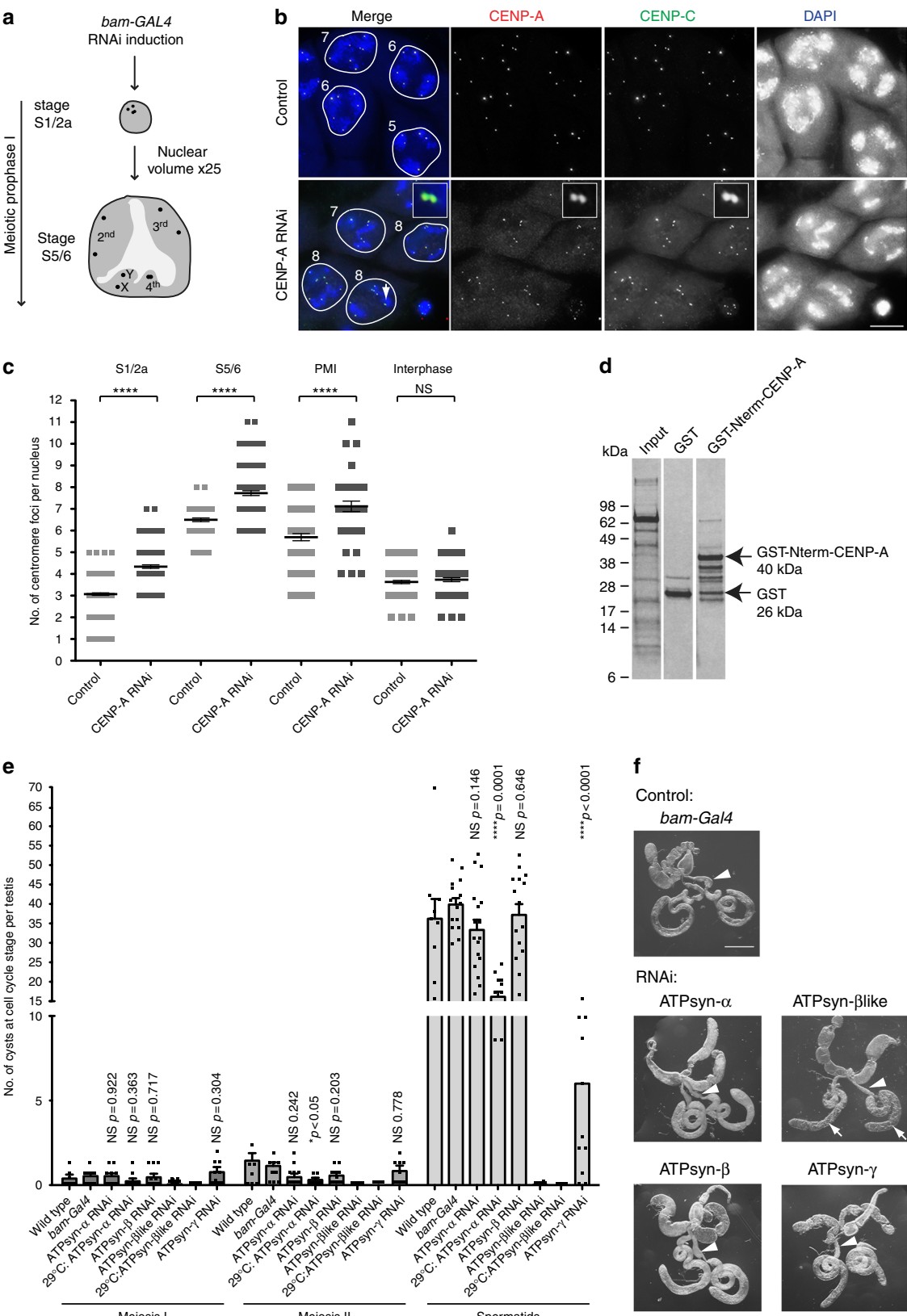

**ATP synthase $F_1$ subunits function in centromere cohesion**. Following cell cycle analyses, we assayed centromere integrity by immunostaining for CENP-A and CENP-C (Fig. 2a). In S5/6 spermatocytes depleted for ATPsyn-α, -βlike and -γ an unexpected increase in the number of centromere foci was observed compared to controls. Quantitation of foci revealed a significant increase upon ATPsyn-α, -βlike and -γ depletion in testes using two independent RNAi lines (****$p < 0.0001$), however no statistically significant increase was measured for ATPsyn-β depleted testes (Fig. 2b). Testes depleted for ATPsyn-βlike or -γ contained spermatocytes with greater than 8, and up to 16, foci per nucleus indicating a premature loss in sister centromere cohesion normally maintained through S6[9,17]. At early prophase I (S1/2a) we also observed differences in the expected number of centromere foci (Fig. 2c). In control nuclei, a mean of 3.072 foci was quantified at S1/2a (Fig. 2d). This number was not significantly different in testes depleted of ATPsyn-βlike ($p = 0.135$), but decreased to 2.7 in testes depleted of ATPsyn-β (*$p = 0.014$) and increased to 3.45 in testes depleted for -α (***$p = 0.0007$) and 4.0 in testes depleted for -γ (****$p < 0.0001$) (Fig. 2d). Strikingly, testes depleted for ATPsyn-βlike arrest at prometaphase I, in line with cell cycle analyses (Fig. 1e).

To assay whether observed meiotic phenotypes might be due to reduced ATP supply, we quantified the level of ATP in testes depleted for ATPsyn-α, -β, -βlike and -γ. Direct measurement of ATP concentration in adult testes confirmed a reduction on average to 45% in ATPsyn-α RNAi (****$p < 0.0001$), 55% in ATPsyn-βlike RNAi (****$p < 0.0001$) and 66% in ATPsyn-γ RNAi (***$p < 0.001$) (Fig. 2e). In ATPsyn-β RNAi samples a modest reduction in ATP level was measured (*$p = 0.044$) and the ATP level was not perturbed in CENP-A RNAi samples ($p = 0.725$). Importantly, we found that the reduction in ATP in RNAi samples did not correlate with phenotype severities (Fig. 2b) or cell cycle delays (Fig. 1e). For example, the ATP level was similarly reduced in testes depleted for ATPsyn-α and ATPsyn-βlike, yet spermatocytes depleted for ATPsyn-βlike display a severe loss in sister centromere cohesion and a cell cycle arrest at prometaphase I, whereas spermatocytes depleted for ATPsyn-α produce mature sperm. To test whether an acute reduction in ATP supply was sufficient to induce excess centromere foci in spermatocytes, we treated wild-type larval testes ex vivo with ATP synthase inhibitors (2,4-Dinitrophenol, oligomycin A) and with an inhibitor of ATP hydrolysis (Sodium Azide ($NaN_3$))[18] and immuno-stained for CENP-A and CENP-C. Quantitation revealed no significant increase in centromere foci at S5/6 after drug treatments, despite an ~70% reduction in ATP (Supplementary Fig. 2C). Finally, knockdown of an additional ATP synthase $F_1$ subunit, ATPsyn-b, as well as ATP synthase complex I components ND23 (NADH

dehydrogenase (ubiquinone) 23 kDa subunit) and ND51 (NADH dehydrogenase (ubiquinone) 51 kDa subunit), did not result in a premature loss-of-sister centromere cohesion at S5/6 despite a comparable reduction in ATP supply (Supplementary Fig. 2D, 2E). Taken together, these results suggest that ATP synthase components -α, -βlike and -γ might function in sister centromere cohesion through a mechanism distinct from canonical roles in ATP generation. To explain how defects in sister chromatid cohesion might result in a prometaphase I arrest/delay, we stained testes depleted for ATPsyn-α or -βlike with antibodies recognising MEI-S332 (*Drosophila* Shugoshin), which localises to and functions at centromeres to protect cohesion at this cell cycle time[19] and may require CENP-A for its localisation[20]. MEI-S332 localised to centromeres at prometaphase I as expected in controls[21] (Fig. 2f). Yet, in 100% of ATPsyn-α-depleted prometaphase I spermatocytes with abnormal nuclei, MEI-S332 did not localise to centromeres and was excluded from the nucleus. Strikingly, in ATPsyn-βlike-depleted prometaphase I arrested spermatocytes, in 100% of cells analysed MEI-S332 at centromeres was reduced and it localised unexpectedly to chromosome arms.

**ATP synthase $F_1$ subunits function in arm cohesion**. We next assayed whether observed defects in sister chromatid cohesion were limited to centromeres. We performed FISH using a probe recognising a non-centromeric heterochromatin site on the second and third chromosome arms (1.686 g/cm³ satellite) (Fig. 3a). Sister chromatid arm cohesion is maintained at this site from S1 to S6[22]. At S1, one to three compact 1.686 signals (on average 2) are normally visible per nucleus and at S5/6, one or two compact 1.686 signals are visible per 2nd/3rd chromosome territory (3–4 signals per nucleus). In testes depleted for ATPsyn-α, -β or -βlike no significant difference in sister chromatid arm cohesion at S1 was observed (Supplementary Fig. 3A, 3B). At S5/6, one or more 1.686 signals per nucleus was frequently less compact in testes depleted for ATPsyn-α, -β, -βlike or -γ (Fig. 3b). Quantitation of 1.686 signals per S5/6 nucleus revealed a significant increase (***$p < 0.001$) in nuclei depleted for ATPsyn-α, -β, -βlike or –γ indicating a loss-of-sister chromatid arm cohesion (Fig. 3c). The disruption of arm cohesion was most pronounced in ATPsyn-βlike-depleted nuclei in which >5 foci per nucleus was frequently observed. We next performed FISH on S5/6 nuclei depleted for ATPsyn-α, -β, -βlike or -γ using a probe recognising a known homologue pairing site (AATAT) on the 4th chromosome (Fig. 3a, d). Quantitation of AATAT signal, which overlaps with a compact DAPI-stained DNA territory, showed that control S5/6 nuclei display either a single AATAT spot (one-spot pattern, 71.69 ± 13.71%) or two associated spots (<5 μm, 21.35 ± 17.04%) or a diffuse spot (6.05 ± 4.59%) (Fig. 3e). Testis-specific

**Fig. 1** Knockdown of CENP-A and ATP synthase $F_1$ subunits in testis. **a** Cartoon of the typical nuclear morphology of meiotic prophase I S1/2a and S5/6 stage spermatocytes showing autosomal and sex chromosome territories (grey) and associated centromeres (black foci). Timing of the *bam*-GAL4 driven RNAi is indicated. **b** Immuno-fluorescent micrograph of control (isogenic) S5/6 nuclei or nuclei RNAi-depleted of CENP-A (at 25 °C) stained with antibodies against CENP-A (red) and CENP-C (green) ($n = 3$). DNA is stained with DAPI (blue). Numbers indicate centromere foci per nucleus; inset shows two spots (indicated by white arrow) typically counted as two individual centromere foci. Scale bar = 10 μm. **c** Quantitation of centromere foci in control nuclei or nuclei RNAi-depleted of CENP-A (at 25 °C) at S1/2a, S5/6, prometaphase (PMI) or interphase stages of meiosis I. Data pooled from two independent experiments, 50 nuclei quantified per experiment. Error bars = SEM. The data were analysed using an unpaired Student's t-test, ****$p < 0.0001$, NS = not significant, $p > 0.05$. **d** Silver-stained SDS-PAGE gel showing input, GST only and GST-Nterm-CENP-A pull-down fractions. Arrows indicate molecular weights of GST and GST-Nterm-CENP-A in kilodaltons (kDa). **e** Quantitation of cysts at respective cell cycle stages (meiosis I, II or spermatids) in wild type (TRiP isogenic) or *bam*-Gal4 control adult testes ($n = 18$) or testes in which ATPsyn-α ($n = 17$), -β ($n = 15$), -βlike ($n = 13$) and –γ ($n = 12$) is RNAi-depleted at 25 °C or 29 °C. Data pooled from two individual RNAi experiments; significance tests were carried out using pooled controls (wild type and *bam*-Gal4). Error bars = SEM. The data were analysed using an unpaired Student's t-test, NS = not significant. **f** Bright-field micrograph of 5 day old control adult testis (*bam*-Gal4) or testis RNAi-depleted of ATPsyn-α, -β, βlike and –γ at 25 °C ($n = 3$). Arrowheads indicate seminal vesicles and arrows indicate abnormal testis morphology. Scale bar = 500 μm

knockdown of ATPsyn-α, -β, -βlike or -γ revealed a general reduction in the one-spot pattern compared to controls and an increase in the two unassociated spot pattern (>5 μm), which was never observed in controls. Finally, the diffuse spot pattern was more frequently observed in spermatocytes depleted for ATPsyn-α (21.92 ± 12.77%), -β (17.94 ± 5.18%), and –γ (16.29%) compared to the control and ATPsyn-βlike RNAi (22.99 ± 24.95%)

showed high variability between experiments. We conclude that reduced expression of ATPsyn-α, -β, -βlike and -γ in testes leads to defects in arm cohesion and 4th homologue pairing or cohesion in prophase I. However, in CENP-A-depleted S5/6 nuclei, we did not observe a defect in sister chromatid arm cohesion (p = 0.5531) (Supplementary Fig. 3C, 3D) or 4th homologue pairing/cohesion (Supplementary Fig. 3E).

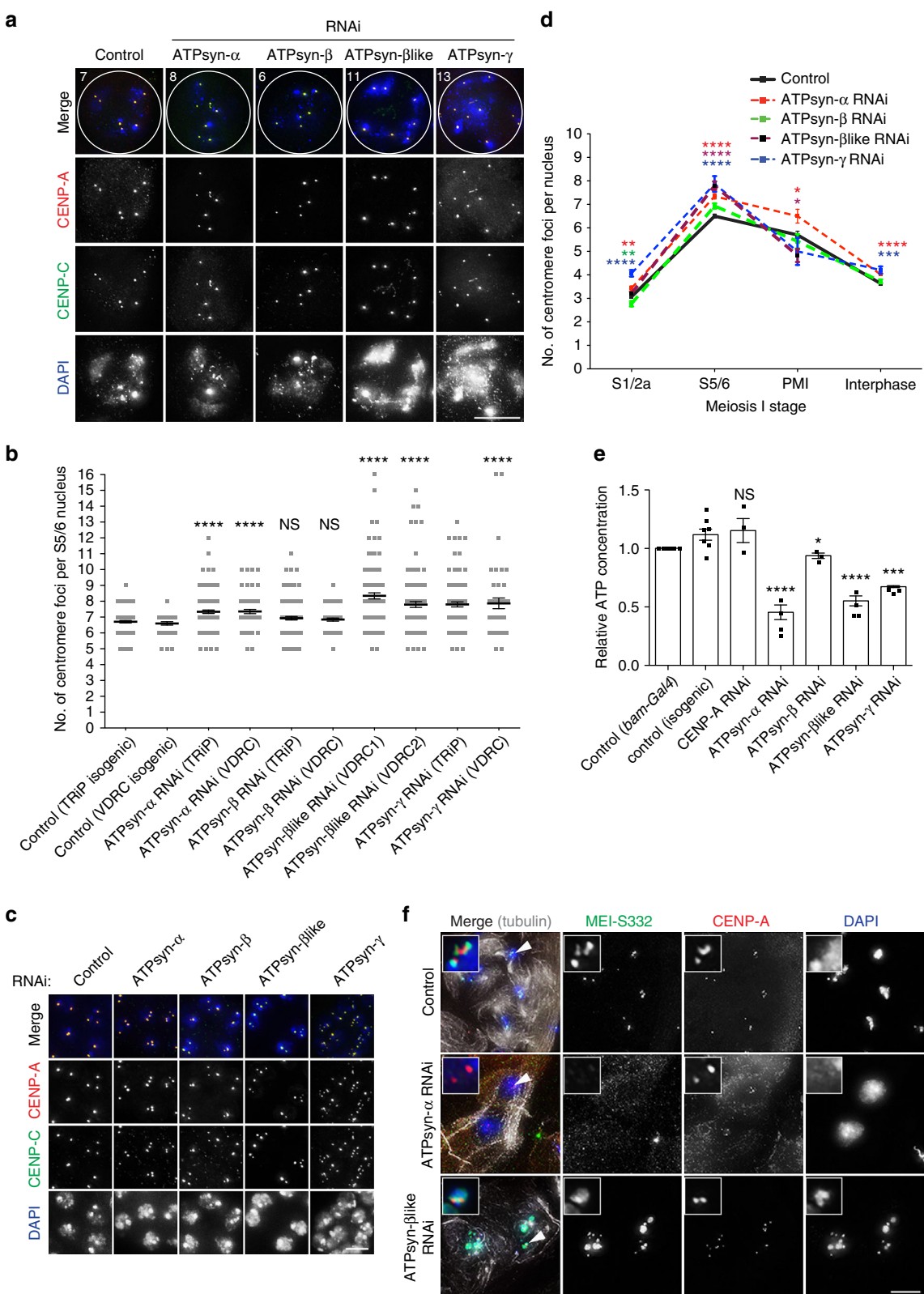

**ATPsyn-α and ATPsyn-βlike localise to centromeres**. We next tested whether CENP-A interacts directly with ATPsyn-α, -β or -βlike. We produced recombinant GST-tagged full length (FL) CENP-A and His-tagged ATPsyn-α, -β and -βlike and performed pull-down interactions in vitro. Western analysis with an anti-His antibody revealed an interaction between GST-FL-CENP-A and His-ATPsyn-α (Fig. 4a). Following this, we investigated the possible nuclear localisation of ATPsyn-α and -βlike using a protocol that extracts the cytoplasm, removing the mitochondrial signal. Surprisingly, using an anti-ATPsyn-α antibody, we found that ATPsyn-α signal partially overlapped with YFP-tagged CENP-C at centromeres in S5/6 nuclei (Fig. 4b). Similarly, GFP-tagged ATPsyn-βlike expressed in vivo under its endogenous promoter partially overlapped with CENP-A at centromeres in S5/6 nuclei (Fig. 4c). To confirm centromeric localisation, we over-expressed GFP-ATPsyn-βlike in cultured *Drosophila* S2 cells and could detect it at centromeres upon cytoplasm extraction (Supplementary Fig. 4A). Re-introduction of GFP-ATPsyn-βlike into flies lacking one copy of ATPsyn-βlike (ATPsyn-βlike$^{+/-}$) that display a defect in sister centromere cohesion, reduced the mean number of centromere foci at S5/6 (Supplementary Fig. 4B). However, this reduction was not significant ($p = 0.084$), indicating only a partial functional rescue by the transgene. To quantify centromeric localisations at S5/6, we calculated a Pearson Coefficient of Correlation of >0.5 for ATPsyn-α and YFP-CENP-C at centromeres ($0.6443 \pm$ sd 0.1030) and a weaker correlation for GFP-ATPsyn-βlike and CENP-A ($0.5473 \pm$ sd 0.0896) (Supplementary Fig. 4C). Antibody staining also confirmed endogenous ATPsyn-α and ATPsyn-βlike overlap in discrete nuclear foci (Supplementary Fig. 4D). In contrast, mCherry-tagged ATPsyn-β was not detectable at centromeres in germ cells (Supplementary Fig. 4E). To test whether CENP-A recruits ATPsyn-α or ATPsyn-βlike to centromeres in vivo we performed CENP-A RNAi at 29 °C and either immuno-stained testes for ATPsyn-α (Fig. 4d) or directly fixed GFP-ATPsyn-βlike signals (Fig. 4e). Costaining with CENP-A or -C confirmed a significant increase in the number of centromere foci per nucleus upon CENP-A RNAi (8.9 or 8.4 foci, ****$p < 0.0001$). In controls, ATPsyn-α and GFP-ATPsyn-βlike localised to centromeres at S5/6 (Fig. 4d, e). However, upon CENP-A depletion, centromeric ATPsyn-α ($n = 46/50$ nuclei) and GFP-ATPsyn-βlike ($n = 47/47$ nuclei) signals were no longer detectable at S5/6.

**CENP-A N terminus promotes sister centromere cohesion.** Finally, to map the interaction site between CENP-A and ATPsyn-α, we immobilised the CENP-A N terminus (residues 1–126) using peptide array-based techniques and probed arrays with recombinant His-ATPsyn-α. Peptide spots corresponding to N terminal conserved sequence blocks B1 and B2 revealed an interaction[23] (Fig. 5a). To test the importance of the CENP-A N terminus in meiosis we generated a fly line expressing a GFP-tagged CENP-A transgene lacking amino acids 1–118 (GFP-CENP-A-Δ118) (Fig. 5b), which removes the B1 and B2 domains but leaves the functional B3 domain intact[24]. As a control, we utilised a line in which GFP is inserted at the identical position, but leaves the N terminus intact (GFP-CENP-A) (Fig. 5b), previously shown to complement lethal *cenp-a* null alleles[25]. Full length GFP-CENP-A expression had a dominant negative effect (*$p = 0.0142$) on the number of centromere foci at S5/6 (6.684 compared to 6.412 in wild type). Truncated GFP-CENP-A-Δ118 localised to centromeres, but showed a dominant negative effect (****$p < 0.0001$) on the number of centromeric foci at S5/6 compared to nuclei expressing full length GFP-CENP-A (Fig. 5c). These results suggest that perturbation of the CENP-A N terminus can disrupt sister centromere cohesion in meiotic prophase I.

## Discussion

Here, in addition to an expected function in ATP synthesis, we report a function for *ATPsyn-α* and *ATPsyn-βlike* in male meiosis and fertility. We show that in testes depleted for ATPsyn-α or –βlike prophase I cells accumulate prior to meiosis I, providing a possible explanation for observed sterility in previously isolated mutants[11,13]. Given that canonical *ATPsyn-β* expression in testis is reduced compared to whole adults[12], *ATPsyn-βlike* might normally compensate for *ATPsyn-β* function. Moreover, although the expression pattern of *ATPsyn-βlike* is entirely consistent with a testis-specific function, we note *ATPsyn-βlike* expression at larval and pupal stages (modENCODE RNA-Seq). This raises the possibility that *ATPsyn-βlike* adopts additional functions in development, which we have not addressed in this study. In addition to its canonical role, we report a nuclear function for ATPsyn-α and -βlike, in particular at centromeres. We find that ATPsyn-α and -βlike localise to centromeres at meiotic prophase I and that this localisation requires CENP-A. Moreover, CENP-A, ATPsyn-α and ATPsyn-βlike are each required to maintain sister centromere cohesion at this stage. Remarkably, ATPsyn-βlike specifies the enrichment of the cohesion protector MEI-S322 to centromeres at prometaphase I, perhaps comparable to the Chromosome Passenger Complex subunit INCENP[26]. In contrast, ATPsyn-α appears to have a distinct function in the nuclear and centromeric localisation of MEI-S332. MEI-S332 mis-localisation to global chromatin in ATPsyn-βlike-depleted nuclei is particularly striking and might be a consequence of a sustained prometaphase I arrest or indicates a more general function of

**Fig. 2** Centromere defects upon ATP synthase-α/-β/-βlike/-γ RNAi. **a** Immuno-fluorescent micrograph of control S5/6 nuclei or nuclei RNAi-depleted of ATPsyn-α, -β, -βlike and –γ (at 25 °C) stained with antibodies against CENP-A (red) and CENP-C (green) ($n = 3$). DNA is stained with DAPI (blue). Numbers indicate centromere foci per nucleus. Scale bar = 10 μm. **b** Quantitation of centromere foci per control S5/6 nucleus or nucleus RNAi-depleted of ATPsyn-α, -β, -βlike and –γ (at 25 °C). For each RNAi sample, *p*-values were calculated compared to respective TRiP or VDRC isogenic controls using an unpaired Student's *t*-test. The data ($n = 100$ nuclei) are pooled from three individual experiments. Error bars = SEM. ****$p < 0.0001$, NS = not significant, $p > 0.05$. **c** Immuno-fluorescent micrograph of control S1/2a nuclei or nuclei RNAi-depleted of ATPsyn-α, -β, -βlike and –γ (at 25 °C) stained with antibodies against CENP-A (red) and CENP-C (green) ($n = 3$). DNA is stained with DAPI (blue). Scale bar = 5 μm. **d** Line graph showing quantitation of the number of centromere foci per control nucleus or nucleus RNAi-depleted of ATPsyn-α, -β, -βlike and –γ at S1/2a, S5/6, prometaphase (PMI) or interphase stages of meiosis I. The data ($n = 100$) are pooled from three individual experiments and was analysed using an unpaired Student's *t*-test, ****$p < 0.0001$, ***$p < 0.001$, **$p < 0.01$ and *$p < 0.05$. Error bars = SEM. **e** Relative ATP concentration in control adult testes (isogenic, *bam-Gal4*) or testes RNAi-depleted (at 25 °C) for CENP-A, ATPsyn-α, -β, -βlike and –γ. *T*-test compares RNAi knockdowns to isogenic control. Experiments were carried out in triplicate and data are pooled from three independent RNAi experiments. Significance was analysed using an unpaired Student's *t*-test, ****$p < 0.0001$, *$p < 0.05$, NS = not significant $p > 0.05$. Error bars = SEM. **f** Immuno-fluorescent micrograph of control nuclei at prometaphase I or perturbed prometaphase I nuclei RNAi-depleted for ATPsyn-α or -βlike stained for MEI-S332 (green), CENP-A (red) and tubulin (grey) ($n = 2$). DNA is stained with DAPI (blue). Scale bar = 10 μm

ATPsyn-βlike on chromatin. In *Drosophila* males, an alternative cohesin complex made up of ORD, SOLO and SUNN maintains meiotic sister centromere cohesion at late prophase I S6[9,17,27]. We find that CENP-A is required for centromere cohesion early in prophase I at S1/2a, prior to ORD, SOLO and SUNN. We suggest

that observed defects in cohesion lead to failed progression through meiosis I and ultimately reduced fertility or sterility. Intriguingly, depletion of ATP synthase $F_1$ subunits also disrupts sister chromatid arm cohesion and 4th homologue pairing/cohesion, suggesting additional global functions outside of the

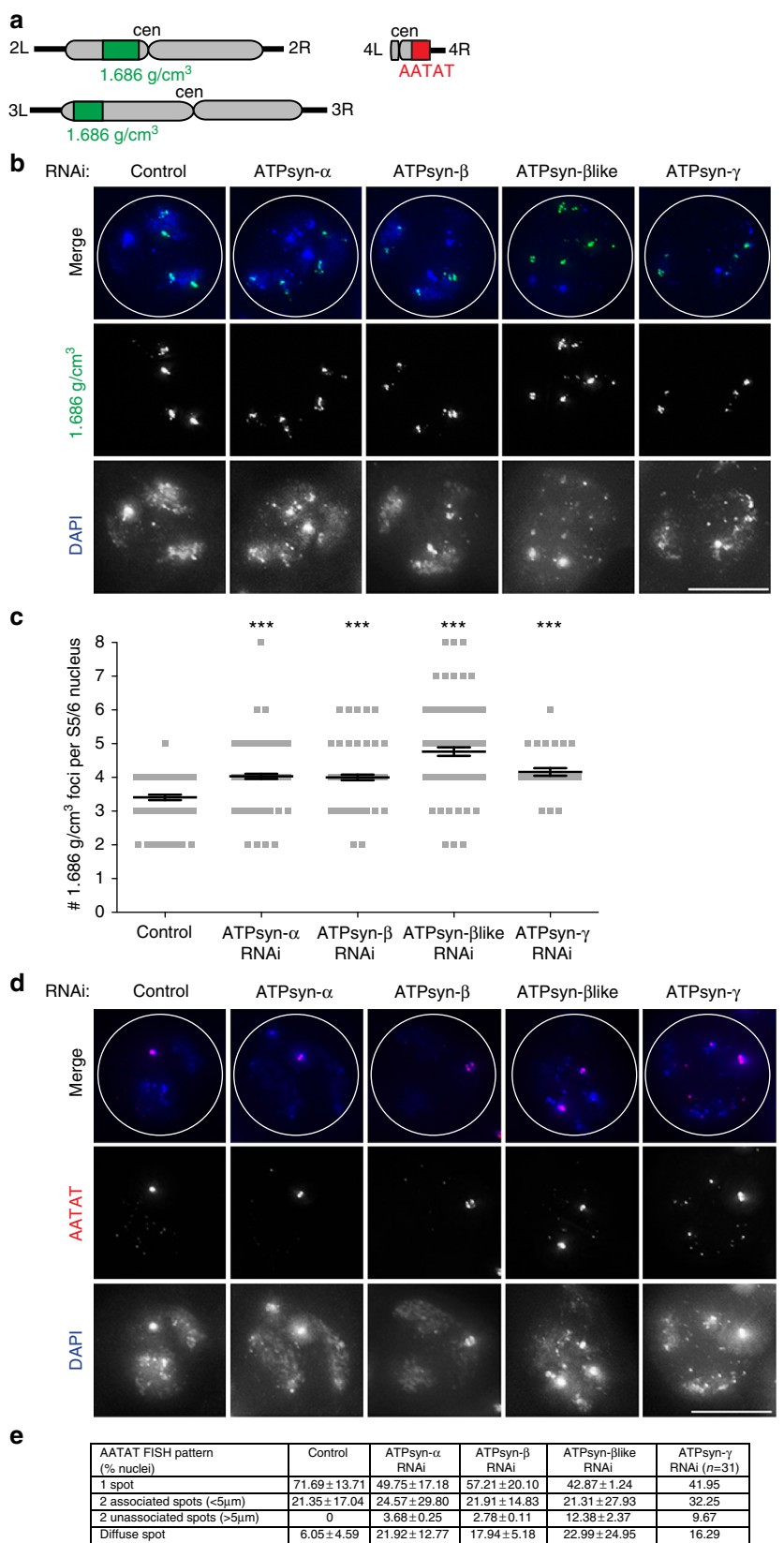

| AATAT FISH pattern (% nuclei) | Control | ATPsyn-α RNAi | ATPsyn-β RNAi | ATPsyn-βlike RNAi | ATPsyn-γ RNAi (n=31) |
|---|---|---|---|---|---|
| 1 spot | 71.69±13.71 | 49.75±17.18 | 57.21±20.10 | 42.87±1.24 | 41.95 |
| 2 associated spots (<5μm) | 21.35±17.04 | 24.57±29.80 | 21.91±14.83 | 21.31±27.93 | 32.25 |
| 2 unassociated spots (>5μm) | 0 | 3.68±0.25 | 2.78±0.11 | 12.38±2.37 | 9.67 |
| Diffuse spot | 6.05±4.59 | 21.92±12.77 | 17.94±5.18 | 22.99±24.95 | 16.29 |

centromere. The ATPsyn-α subunit directly interacts with the CENP-A N terminus, providing a first function for conserved B1 and B2 domains[23]. We propose that ATPsyn-α recruits ATPsyn-βlike to centromeres. Our functional analyses of flies expressing a GFP-tagged CENP-A lacking amino acids 1–118 show the CENP-A N terminus is not required for meiotic centromere localisation, different from plants[3–5]. Instead, the fly CENP-A N terminus appears to be important for meiotic sister centromere cohesion, possibly via the recruitment of ATPsyn-α and ATPsyn-βlike.

Our data support a model in which mitochondrial ATP synthase $F_1$ subunits adopt nuclear functions that appear to be independent of ATP production. First, ATPsyn-α and ATPsyn-βlike interact with CENP-A/centromeres. Second, the severity of observed meiotic phenotypes does not correlate with ATP supply. Third, ATP depletion was not sufficient to induce a loss of cohesion. Finally, our findings are in line with an ATP independent requirement for ATPsyn-α, -β and -γ in germ line stem cell differentiation in *Drosophila* females[16]. In conclusion, we propose that the CENP-A N-terminus recruits ATPsyn-α and ATPsyn-βlike to centromeres to promote sister centromere cohesion in a novel nuclear function that is independent of canonical roles in oxidative phosphorylation.

## Methods

**Fly stocks and husbandry.** Stocks were cultured on standard cornmeal medium (NUTRI-fly) preserved with 0.5% propionic acid and 0.1% Tegosept at 20 °C under a 12 h light dark cycle. UAS-RNAi lines were obtained from the Bloomington Stock Centre, the Transgenic RNAi Project (TRiP) or Vienna *Drosophila* RNAi Centre (VDRC) (Supplementary Table 1). Appropriate isogenic RNAi lines for TRIP or VDRC were used as controls. The testes-specific promoter *bam* was used to drive GAL4 expression (*w;; bam-Gal4-VP16, UAS-dcr2*; provided by M.Fuller) and crosses were performed at 25 °C or 29 °C. Transgenic lines expressing N terminal tagged mCherry-ATPsyn-β, eGFP-ATPsyn-βlike or eGFP-Δ118-CENP-A under respective endogenous promoters were generated by transposable (P) element transformation of pCaSpeR5 vector in $w^{1118}$ embryos (injection, selection and balancing by BestGene Inc). *ATPsyn-βlike* cDNA was amplified from wild type with 900 bp upstream and 600 bp downstream. Δ118 *cid* (bp 354–678) cDNA was amplified from wild type with 413 bp upstream and 417 bp downstream; a 3× glycine linker was placed between the GFP tag and the *cid* start codon. Transgenic flies expressing GFP-CENP-A and YFP-CENP-C were gifts from C. Lehner and S. Heidmann[25]. Flies harbouring an insertion in *ATPsyn-βlike* ($w^{1118}$; *PBac{w[ + mC] = RB}ATPsynbetaLe01800*) were obtained from the Bloomington Stock Centre (17989). For fertility tests, two virgin age-matched males/females were crossed, allowed to lay eggs for 2 days and the number of adult progeny was scored after 20 days. For cell cycle analysis, the number of cysts in meiosis I, II or containing spermatids in at least 12 testes pooled from two individual RNAi experiments from adults <5 h old were scored.

**Recombinant protein production and in vitro binding assays.** GST, GST-Nterm-CENP-A, GST-FL-CENP-A (full-length), His-ATPsyn-α, His-ATPsyn-β and His-ATPsyn-βlike were expressed in BL21 Star™ Codon-Plus-RIL *E. coli*. His-tagged ATPsyn-α, -β, and –βlike were solubilised from inclusion bodies in 5 M urea, purified under denaturing conditions using Ni-NTA HisPur agarose beads and re-natured by stepwise dialysis into 50 mM Tris-HCL, pH 8.0. For tissue protein extracts, wild-type adult testes were digested in 1X PBS containing 1 mg/ml collagenase, 100 µg/ml DNase and 1.5 mM CaCl₂, passed through a 40 µm sieve and treated with a hypotonic buffer (10 mM HEPES, 1.5 mM NaCl, 1.5 mM MgCl₂, 0.1 mM EGTA, 1 mM DTT, 0.1% Triton X-100, 1% protease inhibitor cocktail) before lysis in 300 mM NaCl. For GST pull-down, GST or GST-Nterm-CENP-A (amino acids 1–126) was incubated with testes extracts for 3 h at 4 °C, followed by

the addition of glutathione agarose beads for 1 h. Precipitated proteins were eluted and analysed by silver staining (SilverQuest, Invitrogen). For mass spectrometry (MS), gel lanes were excised and trypsin-digested for analysis by Nano LC-MS/MS (Proteomics Facility, University of Bristol). The CENP-A peptide array of 18-mer overlapping peptides was generated by automatic SPOT synthesis[28] on Whatman 50 cellulose membrane supports using Fmoc (9-fluorenylmethyloxycarbonyl) chemistry with the MultiPep RSi (Intavis Bioanalytical Instruments). Specifically, a library of overlapping peptides 18 amino acids in length, each shifted by four amino acids, and encompassing the sequence of the CENP-A N terminus was SPOT synthesized on nitrocellulose membranes to generate CENP-A arrays[28]. Peptide arrays were challenged with His-ATPsyn-α (5 µg/ml) and binding patterns were revealed by anti-ATPsyn-α western blot.

**IF, FISH and Microscopy.** For Immunofluorescence (IF), testes from young adult males (<1 days old) or 3rd instar larvae were dissected in 1X PBS, gently squashed onto poly-L-lysine coated slides, snap frozen in liquid nitrogen and fixed in 4% paraformaldehyde for 10 min or in cold methanol for 5 min, followed by cold acetone for 2 min (for anti-tubulin staining). For cytosol extraction, samples were immediately washed in 1X PBS-0.1% Triton X-100 (0.1 PBT). For cytosol preservation and for fluorescence in situ hybridisation (FISH), fixed samples were passed through an ethanol series (75–85–95%) at −20 °C and dried prior to permeabilisation in 1X PBS-0.4% Triton X-100 (0.4 PBT) with 0.3% sodium deoxycholate. For IF, samples were blocked in 0.1PBT with 1% BSA for 1 h at room temperature, incubated with primary antibodies overnight at 4 °C and with secondary antibodies for 1 h at room temperature. For FISH, prehybridisation was carried out in 2X Saline Sodium Citrate (SSC)- 0.1% Tween-20 with 50% formamide for 2 h at 37 °C. DNA probes for the 2nd/3rd (AATAACATAG)₃ and 4th (AATAT)₆ chromosomes were directly labelled with Alexa Fluor conjugates (Eurofins). Hybridisation of DNA probes (20 ng) was carried out overnight at 20 °C. Imaging was carried out using a DeltaVision Elite wide-field microscope system (Applied Precision). Images were acquired as z-stacks with a step size of 0.2 µm; raw data files were deconvolved using a maximum intensity algorithm. 3D z-stack images were represented in 2D by projection using SoftWorx (Applied Precision). Focal fluorescent intensities were measured as corrected total cellular fluorescence (CTCF) using Image J software (NIH). Pearson Coefficient of Co-localisation was calculated in SoftWorx.

**ATP assay.** ATP was extracted from 20 adult testes from flies aged 3 to 24 h by homogenisation in a chaotropic buffer and ATP levels were quantified using a luciferase based ATP assay (Molecular Probes) as described[29]. For ATP depletion, testes were treated ex vivo with oligomycin A (50 µg/ml), 2,4-Dinitrophenol (1 mM) or NaN₃ (5 mM) for 1 h.

**Statistical analysis.** To determine the significance, data were analysed using a two-tailed, unpaired *t*-test. NS = $P > 0.05$, *$P < 0.05$, **$P < 0.01$, ***$P < 0.001$ and ****$P < 0.0001$. Normality testes (D'Agostino-Pearson) were carried out in Prism.

**RT-PCR and qPCR.** RNAs were isolated from 100 adult testes (<2 day old) using RNeasy MiniElute kit (Qiagen) and DNase treated. For RT-PCR, 2 µg of RNA was reverse transcribed using SuperScript III Reverse Transcriptase kit (Invitrogen). Quantity of 200 ng of cDNA was used in PCR reactions. Primer sets used for RT-PCR: ATPsyn-βlike: for ATGTTGGTATCATGGGCTAAAATGGCT, rev: TTAATCTTTCTTTTCCGGTTCTTTTGGCTTG, ATPsyn-β: for: ATGTTCGGTTACGTGCTGCA, rev: CTAGGCAGCTTCCTTTGCCAGG. For qPCR, total testes cDNA was produced by reverse transcription of 600 ng of RNA in a 20 µl reaction using Applied Biosystems high capacity RNA-to-cDNA kit. DNA primers with 90–110% efficiencies were selected: ATPsyn-α: for: GGCCCTTAACTTGGAGCCCG, rev: ATGGCACCGGTACGCTTGAC; ATPsyn-β: for: GGTGGCTCTCGTATATGGGC, rev: CGGAAGTACTCTGCAACGGT; ATPsyn-βlike for: GGCCATGGATTCCACCGAAGG, rev: GATGCGTCCCAA AACGGCCT; ATPsyn-γ: for: GATGGTGTCCGCTGCCAAGT, rev: GCCGATG CCGTAAGGACGAG and cenp-a: for: GAAGACGGCCACCGACTACGG, rev: CGTCGAGGAACGCCGATTGT. Quantity of 10 ng of cDNA was used per reaction and qPCR was carried out using PowerUp SYBR Green MasterMix from

**Fig. 3** Arm cohesion defects upon ATP synthase-α/-β/-βlike/-γ RNAi. **a** Cartoon showing chromosomal location of FISH probes used in the study. The 1.686 g/cm³ satellite probe (green) targets both the second (2 L) and third (3 L) chromosome arms. The AATAT repeat probe (red) targets the fourth (4 R) chromosome arm. **b** Micrograph of 1.686 g/cm³ FISH probe (green) performed on control S5/6 nuclei or nuclei RNAi-depleted at 25 °C of ATPsyn-α, -β, -βlike and –γ. DNA is stained with DAPI; white circle outlines the nucleus ($n = 3$). Scale bar = 10 µm. **c** Quantitation of 1.686 g/cm³ foci in control S5/6 nuclei ($n = 81$) or nuclei RNAi-depleted for ATPsyn-α ($n = 122$), -β ($n = 95$), -βlike ($n = 102$) and –γ ($n = 31$). Data pooled from two individual experiments. Significance was determined using an unpaired Student's *t*-test, ***$p < 0.001$. Error bars = SEM. **d** Micrograph of AATAT FISH probe (red) performed on control S5/6 nuclei or nuclei RNAi-depleted of ATPsyn-α, -β, -βlike and -γ. DNA is stained with DAPI; white circle outlines the nucleus ($n = 3$). Scale bar = 10 µm. **e** Quantitation (% nuclei ± SD) of the AATAT hybridisation pattern in control S5/6 nuclei or nuclei RNAi-depleted of ATPsyn-α, -β, -βlike ($n = 2$, 50 nuclei quantified per experiment) and -γ ($n = 31$ nuclei)

Applied Biosystems. qPCR reactions were carried out according to the manufacturers fast cycling specifications on a StepOne Plus real-time PCR system. Three technical replicates were carried out per experiment and data was pooled from two independent RNAi knockdowns. *gapdh* and *rpl32* were used as internal control genes and fold change (ΔΔCt) was calculated using the comparative Ct method[30].

**Antibodies**. IF primary: rabbit anti-CENP-A (Active Motif, #39713, 1:1000); guinea pig anti-CENP-C[31] (1:1000); rabbit anti-ATPsyn-α (Abcam #ab151229, 1:100); mouse anti-ATPsyn-β (Abcam #ab14730, 1:200); goat anti-ATPsyn-γ (Abcam #ab190310, 1:200); rat anti-mCherry (Chromotek 5F8 1:500); mouse anti-tubulin (Sigma DM1A, 1:100); guinea pig anti-MEI-S332[32] (gift from T. Orr-

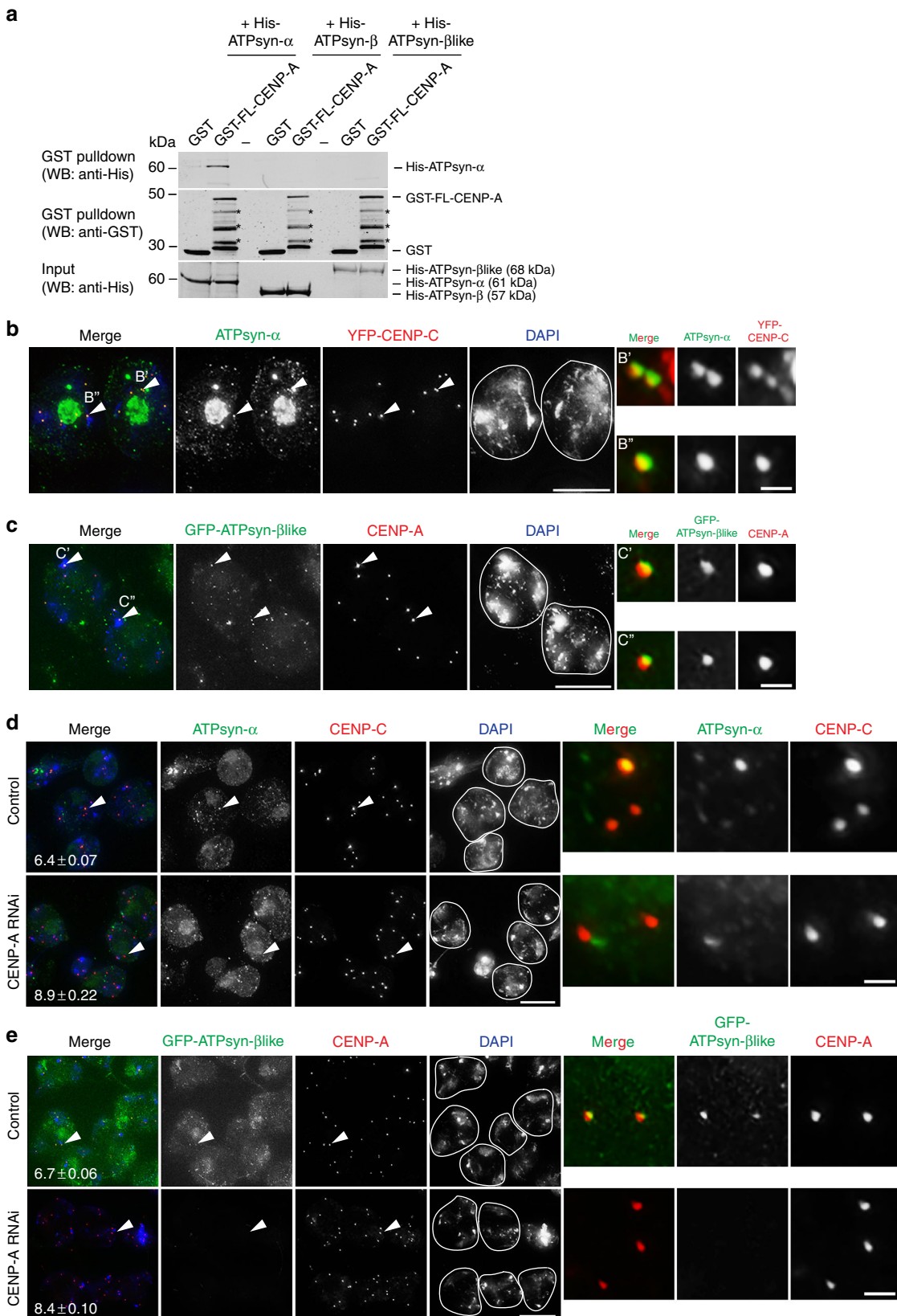

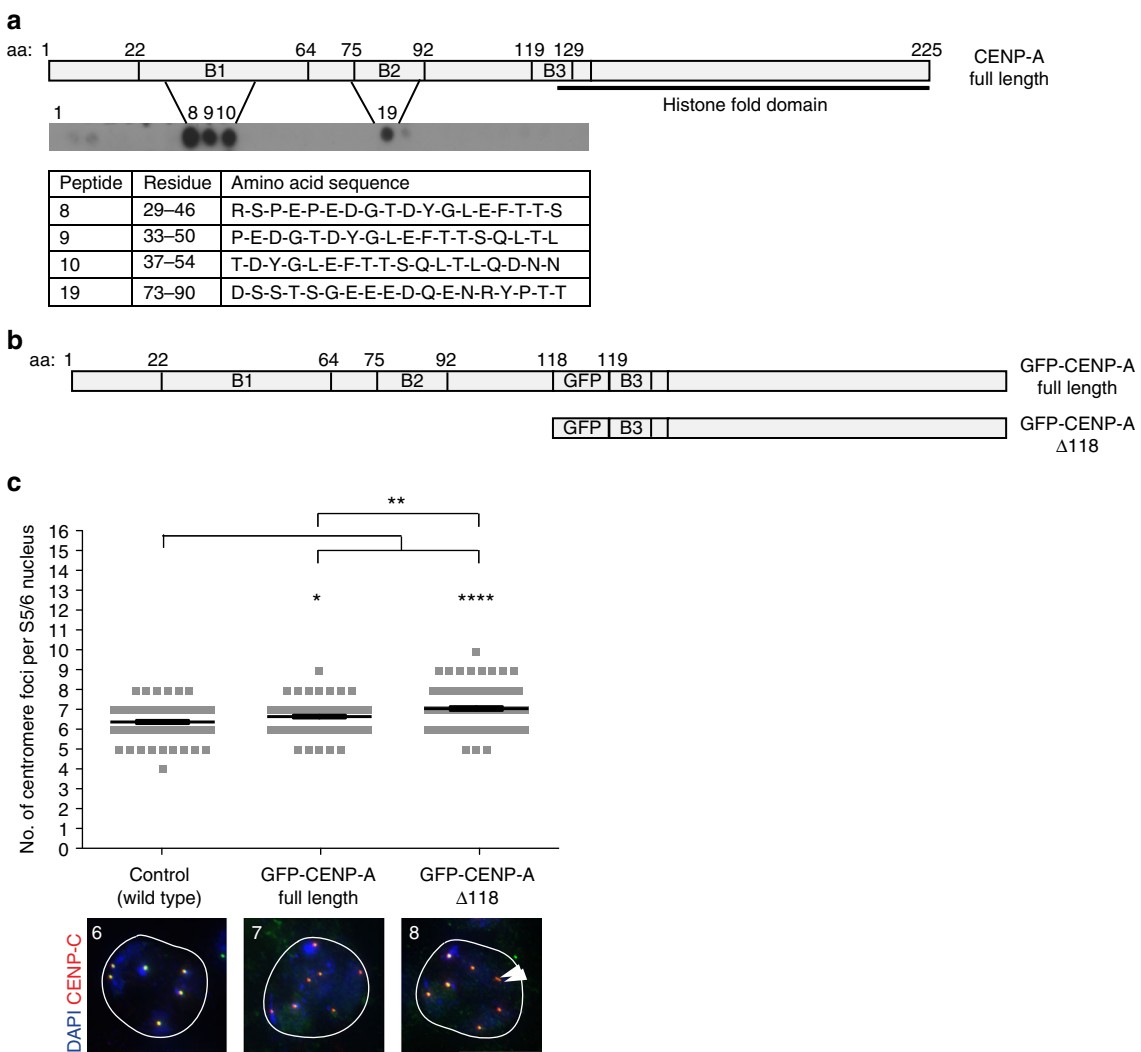

**Fig. 5** Requirement for CENP-A N terminus in centromere cohesion. **a** Peptide arrays encompassing the CENP-A N terminus (amino acids 1–126) probed with His-ATPsyn-α, followed by western analysis with an anti-His antibody ($n = 3$). Schematic of CENP-A N terminus showing position of conserved B1, B2 and B3 domains and histone fold domain. Table displays the amino acid identity of interacting peptides 8, 9, 10 and 19. **b** Schematic showing the position of GFP between amino acid 118 and 119 of CENP-A in GFP-CENP-A full length and GFP-CENP-A-Δ118 transgenes. **c** Top: quantitation of the number of centromere foci per S5/6 nucleus from fly lines expressing GFP-CENP-A or GFP-CENP-A-Δ118 (both homozygous insertions) in addition to endogenous CENP-A. Data pooled from three individual RNAi experiments. Significance was determined using an unpaired Student's t-test, ****$p < 0.0001$, **$p < 0.01$, *$p < 0.05$. Error bars = SEM. Bottom: representative images of S5/6 nuclei immuno-stained for CENP-C (red) and CENP-A or GFP (green). White arrowhead indicates two sister centromeres. Number of centromere foci per nucleus is marked. Scale bar = 10 μm

**Fig. 4** ATPsyn-α and ATPsyn-βlike localisation at centromeres. **a** In vitro pull-down interaction with full-length GST-tagged CENP-A (GST-FL-CENP-A) and His-tagged ATPsyn-α, -β or –βlike revealed by western analysis with an anti-His antibody ($n = 3$). *GST-FL-CENP-A cleavage products. **b** Immuno-fluorescent micrograph of prophase S5/6 nuclei from flies expressing YFP-CENP-C (red) co-stained with antibodies against ATPsyn-α (green). DNA is stained with DAPI (blue). Co-localisation was observed in three independent experiments. Scale bar = 15 μm. B′ and B″ indicate centromeres enlarged in inset. Scale bar = 1 μm **c** Immuno-fluorescent micrograph of prophase S5/6 nuclei from flies expressing GFP-ATPsyn-βlike (green) co-stained with antibodies against CENP-A (red). DNA is stained with DAPI (blue). Co-localisation was observed in three independent experiments. Scale bar = 15 μm. C′ and C″ indicate centromeres enlarged in inset. Scale bar = 1 μm. **d** Immuno-fluorescent micrograph of adult testis in which CENP-A was RNAi-depleted at 29 °C or controls (TRIP isogenic) stained with antibodies against ATPsyn-α (green) and CENP-C (red) ($n = 3$). Scale bar = 15 μm. Arrowheads indicate centromeres enlarged in inset. Scale bar = 1 μm. Average number of centromere foci per nucleus ± SEM is marked ($n = 100$ nuclei, pooled from two experiments). **e** Immuno-fluorescent micrograph of adult testis from flies expressing GFP-ATPsyn-βlike (green) in which CENP-A was RNAi-depleted at 29 °C or sibling controls stained with antibodies against CENP-A (red) ($n = 3$). Scale bar = 15 μm. Arrowheads indicate centromeres enlarged in inset. Scale bar = 1 μm. Average number of centromere foci per nucleus ± SEM is marked ($n = 100$ nuclei, pooled from two experiments)

Weaver 1:500) and rabbit anti-GFP (Santa Cruz SC-8334). Guinea pig anti-ATPsyn-βlike antibodies (1:200) were generated by co-injection of two KLH and BSA conjugated peptides CKTDAELVKKKDE (amino acid 68–80) and GDAP-PAKAEAKKDEK (amino acid 575–587) marked on Supplementary Fig. 1D. IF secondary: Alexa-488, -546, -647-coupled goat anti-mouse, goat anti-rabbit or goat anti-guinea pig (Life Technologies, 1:500). Western analysis: rabbit anti-CENP-A (Active Motif, #39713, 1:1000), mouse anti-ATPsyn-α (Abcam #ab14748, 1:1000), mouse anti-ATPsyn-β (Abcam #14730, 1:1000), rat anti-GST (Chromotek 6G9, 1:1000), mouse anti-red (Chromotek 6G6, 1:1000), mouse anti-poly-his (Sigma Aldrich #H1029, 1:1000), mouse anti-tubulin (Sigma Aldrich #T5168, 1:10,000) and rabbit anti-histone 3 (Millipore 17–10,254; 1:50,000); guinea pig anti-ATPsyn-βlike (1:1000). Uncropped scans of blots are provided in Supplementary Figure 5.

**Data availability**. The authors declare that all data supporting the findings of this study are available within the article and its supplementary information files or from the corresponding author upon reasonable request.

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

## Acknowledgements

E.M.D. is funded by SFI-HRB-WT 100105/Z/12/Z and SFI-PIYRA 13/YI/2187. C.M.C. is funded by SFI-PIYRA and NUIG College of Science Studentship. P.K. and B.M. are funded by SFI Grant 12/RI/2345 and 13/CDA/2228. The authors acknowledge the facilities and technical assistance of the NCBES qPCR Facility, funded by NUIG and the Irish Government's Programme for Research in Third Level Institutions, Cycles 4 and 5, National Development Plan 2007-2013. Stocks obtained from the Bloomington Drosophila Stock Center (NIH P40OD018537) were used in this study.

## Author contributions

C.M.C., C.B. and E.M.D. performed *Drosophila* experiments. P.A.K. and B.M. generated CENP-A peptide arrays. E.M.D. and C.M.C. wrote the manuscript.

## Additional information

**Competing interests:** The authors declare no competing interests.

