## [Peer Review File · Nature Communications]

Reviewers' comments:

Reviewer #1 (Remarks to the Author):

The centromere plays an important role in chromosome functions that are specific to meiosis, and understanding these connections is central to advancing knowledge about chromosome inheritance through the germline. Certainly, this topic is the focus of many important ongoing research programs worldwide. There is a longstanding proposal that at least some of these roles include the special chromatin formed at centromeres, including the nucleosomes assembled with the histone H3 variant, CENP-A. In their study, Collins et al propose a role for the ATP synthase complex V in maintaining male germ line sister chromatid cohesion through interactions with centromere specific histone, CENP-A. Further, they report that this function of ATP synthase is independent of its enzymatic activity in oxidative phosphorylation. The authors conclude that subunits of this complex have a role in early prophase cohesion maintenance and that this is coordinated by recruitment to the centromeres through the N-terminus of CENP-A. My excitement for the study comes from its novel advances on a very important and timely area of chromosome segregation research. There is interesting headway made in this paper in a biological process where there are substantial experimental hurdles to overcome. My enthusiasm for the paper, at least in its present form, is tempered by concerns about whether the data substantiates some of the major claims of the paper (these concerns are spelled out, below; major concerns #1-4) and about an inconsistent presentation of the data that makes it confusing to assess (these concerns are spelled out, below; major concerns #5-7). I also list some minor concerns that I envision will be simple to address. Major concerns:

1. The claims of centromere localization of ATP-syn-beta-like in Figure 4 is a critical point in the paper (e.g. it's in the title), but is not compelling as it stands. I understand that there could be a substantial non-centromeric pool, but the overlap with centromeres is difficult to see in the images and the numbers in the quantitation given do not instill much faith that this is meaningful co-localization. It looks to me that its lower expression might just score better in their quantitation scheme. There are methods, each with their own pros/cons, to detect specific interactions at a sub-cellular location that rely on proximity of two components. Some experiment along those lines is needed to substantiate the claims that ATP-syn-beta-like is present at the chromosomal location - the centromere - where the authors propose it is functioning in meiosis.
2. Fig. S1A and 1B: I was surprised that the authors are stuck trying to decipher small changes in centromere foci number with only a 30 % reduction in CENP-A levels. Raychaudhuri et al 2012 reported the GD4436 line gave a reduction of half or more. There is also a VALIUM 20 cid (Fly Stock #40912) RNAi line available that is a short hairpin and may be more effective for this purpose. The authors should explain why they settled on this level of knockdown. Would the effects on centromere foci number be more pronounced with a greater level of knockdown?
3. The authors' claim, that there are ATP independent functions of this complex, is largely based on negative results in RNAi experiments. But in those experiments there is no phenotype, so how is one to conclude that any substantial knockdown occurred at all? Some clear validation (e.g. qRT-PCR and/or immunofluorescence) that the specific RNAi's actually work would help in substantiating this claim. This is vital since the other supporting evidence, that ATPsyn- α has mature sperm but ATPsyn- β like has no spermatids although both have equal reduction in ATP production, could simply be a difference in the amount of knockdown. Such a difference, or not, is not apparent from the immunofluorescence results presented in the paper.
4. This paper presents a central claim of a role of ATP synthase and CENP-A in a specific nuclear function in meiosis, but the model for how this could work is not clear. Sister centromere cohesion normally does not reload during prometaphase, although there is a dynamic component to it. Therefore; in the case of all the RNAi's examined, it is surprising that the centromere cohesion defect is resolved at interphase. This confusing aspect is not discussed by the authors at all. Is it possible that the centromeres are simply clustered at interphase and hence quantification of foci is confounded? In the ATPsyn- α RNAi that actually make sperm, it may be worthwhile to test whether or not meiosis II is affected. Or at least comment on this. The other related aspect, which is not explained is how arm cohesion is affected by centromeric ATP synthase population. The mis-

localization of Shugoshin does not completely explain this result. In addition, there is no mechanism suggested for Shugoshin delocalization to the arms. Do the authors think there may be a disruption in CPC or Bub1 function as reported earlier in Boyarchuk et al 2007? Overall the mechanism of how this complex works with CENP-A is unclear, especially as the ORD/SUNN/SOLO complex is responsible for the same function later in prophase. This is all critical for one to assess the claims of the paper (e.g. the title of the paper): without further explanation, the data do not correlate a functional consequence of early prophase separation with fertility issues per se, especially if the separation problem reverts to a normal state prior to meiosis II.

5. There is a discrepancy between Figures 1C and 2D data points for S1/2a nuclei. The average is defined as 3.6 for control in 1C whereas it is 2.7 in 2D. Hence in 2D, an average of 3.4 foci is considered to be "high" when it is in wild type range in Figure 1. Although I understand that there is variability in RNAi experiments, the control CENP-A foci range should be consistent. The authors do not address this point in the text but it calls into question quantification of the other data points. It might be better to pool all the independent control experiments, define an average for S1/2a and all other stages and then compare all the mutants to the same range.

6. Quantifications: There is a lack of the N used in experiments, and would be helpful in either figure legends or the main text, to list this alongside the P values. Some examples: Figure 1F needs an N. Figure 2E needs clarification of whether 20 testes were used in 1 experiment or 2 independent experiments. To be convincing, Fig. S2A requires numbers in terms of number of cells observed and whether the KD was observed in every cell or in a certain percentage, especially since the rest of the findings are hinged upon these KDs being consistent.

7. Regarding the issues with centromeric localization of ATPsyn- β like (point #1, above), Fig. S4 D can be shown as single channel and with insets zoomed onto the centromeres. Again, an N is required for this panel to be convincing and maybe more pictures to support the claim. Figure 4D inset, should also be present in the single channels to be more convincing since the bright CENP-A staining makes it harder to see the faint GFP- ATPsyn- β like.

Minor concerns:

8. Fig S1E is referred to as S1F (line 119).

9. Are the authors confident that the ATPsyn- β like antibody does not recognize the ATPsyn-B? Some comment in the paper on this possibility should be made.

10. Authors make a strong claim that the N-terminus truncation has a very significant dominant negative effect and while it definitely has an effect, the numbers themselves do not look convincing especially if this domain is correlated with ATPsyn- β like function. Perhaps soften this claim, or test this notion further to strengthen it. For instance, test if this truncation can rescue CENP-A RNAi sister centromere cohesion defect while also measuring centromeric ATP-synB like levels.

11. Table S1 is not explained.

12. Line 259: ATPsyn- β like localization was not shown in mitochondria?

13. Authors show that ATP-synB is expressed in testes Fig S1D, but in line 102, it says that this protein is repressed in testes. Is this stage specific repression or only germ cell specific?

14. Fig 5A shows that ATPsyn- α has the strongest interaction with CENP-A even though in vivo staining does not show any nuclear ATP-syn α . Can the authors detect ATPsyn- α in germ cell nuclei at all? Is it reduced at centromeres in S5/6 in CENP-A and ATPsyn- α RNAi males?

15. To be "nomenclature inclusive", consider also providing the Drosophila world-centric names of the genes in parentheses in Table S2.

Reviewer #2 (Remarks to the Author):

The authors propose a novel function for the beta subunit of the ATP synthase in meiosis. If reproducible, this is an exciting discovery and should be of interest to a wide audience. The authors describe this meiosis function most strongly for a variant of ATP synthase beta called ATP syn beta-like. The manuscript is thorough but there are some deficiencies which need to be addressed before the data are acceptable for publication. In particular, the function of betalike in

meiosis, that is to bind together CENP-A and MEI-S332 and the co-localization of betalike with CENP-A are preliminary and need further investigation. This may not even be the correct function of betalike, as outlined below, given the data the authors present.

1. Figure 2E and suppl data S2B: Although the drugs 2,4-DNP and oligomycin prevent ATP synthesis in the intact mitochondria, it would be important to rule out the role of ATP hydrolysis (of the isolated F1 enzyme) in meiosis. This can be shown with other drugs that block ATP hydrolysis by the beta subunit-containing ATPase, even in the absence of the proton pumping membrane components of the ATP synthase.
2. Figure 4A: The diffuse staining in Figure 4A of MEI-S332 does not demonstrate a lack of binding between MEI-S332 and CENP-A, it just shows a more diffuse localization of MEI-S332, but MEI-S332 is still present at the sites where CENP-A is localized (in the figure shown). To show there is a lack of binding between the two proteins in the absence of betalike, a co-IP needs to be performed.
3. Figure 4B: The background subtraction images need to be shown and the details of the colocalization need to be better illustrated in the figure. Showing the diffuse staining for betalike is misleading and not parallel to the bar graph in 4C.
4. 4D: Same as for 4C, the background subtracted images need to be shown. It is also not clear comparing figures 4B and D why the measurement of fluorescent intensity of betalike co-localizing with CENPA is used rather than the number or foci. The type of measurement used for the other figures (number of foci) should also be shown. What about using a nuclear localization signal to eliminate the cytosolic localization of betalike and show that it is reduced in the CENP-A RNAi?
5. Figure 5A: It is disturbing that the binding between betalike and CENP-A is not direct, particularly since it is betalike reduction that really stops meiosis at early stages in figure 1E, much more than alpha. Therefore the mechanism of binding of MEI-S332 and CENP-A to each other depending on betalike does not make sense. To show the proposed role for betalike better, a mutation should be made in betalike that specifically prevents its co-IP with alpha (since that is the proposed mechanism the authors prefer). This mutant should disrupt binding of MEI-S332 and CENP-A (and co-localization at the centromere) and should halt meiosis at stage I as in Figure 1E. This still leaves the question open, however of why RNAi of alpha does not prevent the early stages of meiosis since by the authors reckoning, it is downstream of betalike in producing the binding between MEI-S332 and CENP-A. Could there be another binding partner or function of betalike that is more important for early meiosis?

Reviewer #3 (Remarks to the Author):

Overall the paper contains some novel findings, and it is very interesting to see a role for ATP synthase subunits in sister centromere cohesion in meiosis, and a striking meiotic phenotype for depletion of the testis-specific ATPsynbeta isoform. Unfortunately there is insufficient detail in many places in the manuscript (methods, numbers for quantitative experiments, magnification/resolution for microscopy) for the reader to be sure that all the findings are robust. The detailed notes below are in order within the manuscript, not in order of priority.

line 80 states 1-3 cenp-A spots visible. line 84 states 3.4 dots in control. Which is correct?
line 88, "problems with sister centromere cohesion are later resolved". What plausible mechanism is there for this? It would be very interesting to follow up on the analysis of numbers of foci and loss of cohesion with a genetic analysis of chromosome segregation defects (for conditions that give some fertility). At minimum you would expect to see aneuploidy in spermatids for those that progress through meiosis - this is easily scored by phase contrast analysis of nuclear size in early spermatids.

line 135, need to state in the text how the depletion was validated.

Figure 1 E, error bars should be SD, not SEM, n (of both nuclei and number of testes) should be stated somewhere. The data for Fig 1C should be shown in the same way as that for Figure 2B (in

which case error bars are not needed at all).

Figure 1D, can you indicated which of the co-purifying bands are which protein.

Figure 1F clearly shows abnormal testis for the RNAi, but a phase contrast image as well is needed to show the arrest at prometaphase 1.

Figure S1C, how can you have a cross-reacting band in the RT-PCR? What is it?

Figure S4D, need to see unmerged images at higher zoom to convince me of the overlap between CENP-A and ATPsynbetalike. As presented there are so many ATPsynbetalike dots that it is almost inevitable that some will overlap the CENP-A dots.

Line 217 "Importantly, GFP-ATPsyn- β like re-introduction into flies lacking one copy of ATPsyn-218 β like partially restores sister centromere cohesion at S5/6 (Figure S4C). But you have not mentioned at all in the main text the evidence that ATPsyn-betalike heterozygotes have any defects to be rescued. The figure legend contains insufficient information regarding the actual numbers of foci per nucleus. Given the very subtle effect here more detail is needed. Is there a significant difference (as expected) between plus and minus rescue construct (only the differences to control are given p-values).

Figure 4B. The quantitation of centromeric ATPsyn-betalike is not at all compelling. The entire testis is full of dots. Is this gene really expressed in all cells in testes (including the hub?), most testis-specifically expressed paralogues of ubiquitously expressed genes are expressed exclusively in the germline. RNA in situ hybridization would reveal which cells the gene is expressed in. Higher resolution microscopy is needed to convince me of the co-localisation.

Figure 5A, there is an error in the labeling of this figure. I assume the first panel is pull down GST rather than pull down anti-his.

Line 250. Avoid using the phrase "more significant". More dramatic maybe, but give us the actual numbers of centromere foci, and the number of nuclei assayed, because the effect from the wild-type tagged construct is subtle. Can the N-terminal deletion cenpA line rescue the lethality of the null allele?

methods. Do not contain sufficient information that the experiments could be repeated. more detail needed for the transgenic constructs. What genomic region was used for each - ie how much flanking dna etc. Were the tags at the c- or n-terminus. what vector was used for the transgenesis - p or site specific recombination?

the p-insertion line for atp-syn-beta-like should be referred to by its correct identifier, not Bloomington stock number.

how long were flies left to lay eggs in the fertility test. Were all flies age-matched.

for protein prep from testes, what is in the hypotonic buffer?

What was GST conjugated to for the pull down? how exactly was the pull down done.

What type of digest was done before MS (trypsin?).

How long was the fixation step for immunostaining?

stats - t tests were used. Were any tests used to assess if the data are normally distributed?

RT-PCR, much more detail needed - how many testes, how much RNA used for each assay. Was DNase treatment used? How many PCR cycles? PCR primer sequences needed. Control genes?

Scoring criteria for determining if two close together spots represent one or two foci, to reduce subjectivity. Is there a minimum size at which a dot counts as a focus? Was the counting done blind?

typos and minor changes.

figure S1 legend, colons not semicolons in B.

Figure S2. These are whole spermatocytes, not just nuclei.

line 174, sites not site for the satellite location (on two chromosomes).

POINT BY POINT RESPONSE TO REVIEWERS COMMENTS

Reviewer #1 (Remarks to the Author):

The centromere plays an important role in chromosome functions that are specific to meiosis, and understanding these connections is central to advancing knowledge about chromosome inheritance through the germline. Certainly, this topic is the focus of many important ongoing research programs worldwide. There is a longstanding proposal that at least some of these roles include the special chromatin formed at centromeres, including the nucleosomes assembled with the histone H3 variant, CENP-A. In their study, Collins et al propose a role for the ATP synthase complex V in maintaining male germ line sister chromatid cohesion through interactions with centromere specific histone, CENP-A. Further, they report that this function of ATP synthase is independent of its enzymatic activity in oxidative phosphorylation. The authors conclude that subunits of this complex have a role in early prophase cohesion maintenance and that this is coordinated by recruitment to the centromeres through the N-terminus of CENP-A. My excitement for the study comes from its novel advances on a very important and timely area of chromosome segregation research. There is interesting headway made in this paper in a biological process where there are substantial experimental hurdles to overcome. My enthusiasm for the paper, at least in its present form, is tempered by concerns about whether the data substantiates some of the major claims of the paper (these concerns are spelled out, below; major concerns #1-4) and about an inconsistent presentation of the data that makes it confusing to assess (these concerns are spelled out, below; major concerns #5-7). I also list some minor concerns that I envision will be simple to address.

Major concerns:

1. The claims of centromere localization of ATP-syn-beta-like in Figure 4 is a critical point in the paper (e.g. it's in the title), but is not compelling as it stands. I understand that there could be a substantial non-centromeric pool, but the overlap with centromeres is difficult to see in the images and the numbers in the quantitation given do not instill much faith that this is meaningful co-localization. It looks to me that its lower expression might just score better in their quantitation scheme. There are methods, each with their own pros/cons, to detect specific interactions at a sub-cellular location that rely on proximity of two components. Some experiment along those lines is needed to substantiate the claims that ATP-syn-beta-like is present at the chromosomal location – the centromere - where the authors propose it is functioning in meiosis.

To support the localisation of ATPsyn-βlike to centromeres, we focus on prophase I spermatocytes expressing GFP-ATPsyn-βlike that have been fixed and costained with an anti-CENP-A antibody (labelled in the far red channel) (Figure 4C). We include a zoom inset of each separate channel as well as the merged channels at high resolution which shows the partial overlap of red and green signals. We have calculated an average Pearson colocalisation co-efficient for GFP-ATPsyn-βlike and CENP-A foci at prophase I S6 stage of $0.5473 \pm \text{sd } 0.0896$ per nucleus ($n=20$ nuclei) (Figure S4C). A value of >0.5 indicates colocalisation. We believe that values are very close to 0.5 as GFP-ATPsyn-βlike and CENP-A do not perfectly colocalise and the signals only partially overlap (foci are green, red and yellow).

In addition we present new data showing the colocalisation of ATPsyn-α (using an anti-ATPsyn-α antibody) and YFP-CENP-C (Figure 4B) or anti-CENP-C (Figure 4D) at centromeres. We have calculated an average Pearson colocalisation co-efficient for ATPsyn-α and YFP-CENP-C foci at prophase I S6 stage of $0.6443 \pm \text{sd } 0.1030$ per nucleus ($n=20$ nuclei) (Figure S4C). We believe that values are close to 0.5 as ATPsyn-α and CENP-C do not perfectly colocalise and the signals only partially overlap (foci are green, red and yellow).

2. Fig. S1A and 1B: I was surprised that the authors are stuck trying to decipher small changes in centromere foci number with only a 30 % reduction in CENP-A levels. Raychaudhuri et al 2012 reported the GD4436 line gave a reduction of half or more. There is also a VALIUM 20 cid (Fly Stock #40912) RNAi line available that is a short hairpin and may be more effective for this purpose. The authors should explain why they settled on this level of knockdown. Would the effects on centromere foci number be more pronounced with a greater level of knockdown?

We have now performed the CENP-A RNAi at 29°C (Vienna Fly Stock v102090) and confirm an almost 50% reduction in CENP-A at centromere at S6 stage (n=30 nuclei) (Figure S1A). We confirm that a higher level of CENP-A knockdown leads to a more pronounced defect in centromere cohesion (average 9.1 foci, n=100 nuclei, p<0.0001) (Figure S4B, S4C). This data is included in Supp Figure 1. In addition, CENP-A RNAi experiments shown in Figure 4D and 4E were performed at 29°C.

3. The authors' claim, that there are ATP independent functions of this complex, is largely based on negative results in RNAi experiments. But in those experiments there is no phenotype, so how is one to conclude that any substantial knockdown occurred at all? Some clear validation (e.g. qRT-PCR and/or immunofluorescence) that the specific RNAi's actually work would help in substantiating this claim. This is vital since the other supporting evidence, that ATPsyn- α has mature sperm but ATPsyn- β like has no spermatids although both have equal reduction in ATP production, could simply be a difference in the amount of knockdown. Such a difference, or not, is not apparent from the immunofluorescence results presented in the paper.

We confirm knockdown of ATPsyn- α , - β , - β like, - γ and CENP-A (cid) at 25°C using qRT-PCR (Figure S1G). We note an almost 4 fold reduction in ATPsyn- α transcript compared to an approximate 10 fold reduction in ATPsyn- β like transcript, which might explain differences in the severity of phenotypes observed. In our cell cycle analyses (Figure 1E), we noted that while testes depleted for ATPsyn- α produce mature sperm, a number of abnormal cysts were observed at the MI and MII stages. To address this concern, we have performed the knockdown and cell cycle analysis of ATPsyn- α and - β like at 29°C (Figure 1E) and we show a representative image of perturbed 16-cell cysts (Figure S2B). Spermatocytes lacking ATPsyn- β like arrest at prometaphase I, whereas many spermatocytes lacking ATPsyn- α have perturbed nuclear morphology at prometaphase I yet proceed through meiosis I resulting in abnormal segregation and can produce mature sperm.

To further confirm knockdown efficiencies, we have included a quantitation of the efficiency of ATPsyn- α (90%), - β (100%), - β like (100%), - γ (100%) knockdown in S6 cells by IF staining with respective antibodies (presented in Supp Fig S2) at 25°C. Additionally, we confirm 100% knockdown of ATPsyn- α and - β like at 29°C (Figure S2B).

4. This paper presents a central claim of a role of ATP synthase and CENP-A in a specific nuclear function in meiosis, but the model for how this could work is not clear. Sister centromere cohesion normally does not reload during prometaphase, although there is a dynamic component to it. Therefore; in the case of all the RNAi's examined, it is surprising that the centromere cohesion defect is resolved at interphase. This confusing aspect is not discussed by the authors at all. Is it possible that the centromeres are simply clustered at interphase and hence quantification of foci is confounded? In the ATPsyn- α RNAi that actually make sperm, it may be worthwhile to test whether or not meiosis II is affected. Or at least comment on this.

To explain how centromere cohesion is resolved at interphase, we propose that SUNN/ORD/SOLO might function to maintain cohesion at this cell cycle stage. It is also possible that the nuclei we quantified at interphase had a sufficient level of CENP-A to progress through meiosis I. We have added the following line to the Results:

The number of centromere foci detected per nucleus at interphase did not differ from the control (3.7 compared to 3.6, p=0.366), possibly due to compensation by additional factors that maintain cohesion at this time or that the CENP-A depletion was less efficient in these cells.

And to the Discussion: *We suggest that observed defects in cohesion lead to failed progression through meiosis I, and ultimately in reduced fertility or sterility.*

By qRT-PCR (Figure S1G), we note an almost 4 fold reduction in ATPsyn- α transcript compared to an approximate 10 fold reduction in ATPsyn- β like transcript, which might explain differences in the severity of phenotypes observed. We also clarify that while testes depleted for ATPsyn- α produced mature sperm, a number of cysts have nuclei with perturbed morphologies at the MI and MII stages. To further confirm or enhance this phenotype, we have performed the knockdown of ATPsyn- α at 29°C, performed a cell cycle analysis (Figure 2E) and show a representative image of an irregular MI cyst (Figure S2B).

The other related aspect, which is not explained is how arm cohesion is affected by centromeric ATP synthase population. The mis-localization of Shugoshin does not completely explain this result. In addition, there is no mechanism suggested for Shugoshin delocalization to the arms. Do the authors think there may be a disruption in CPC or Bub1 function as reported earlier in Boyarchuk et al 2007?

To understand better effects of ATPsyn- β like on the recruitment and localisation of MEI-S332 to centromeres, we investigated in parallel MEI-S332 localisation in ATPsyn- α RNAi. We find that ATPsyn- α and ATPsyn- β like have different effects on MEI-S332 enrichment at centromeres. Knockdown of ATPsyn- β like results in the mislocalisation of MEI-S332 to chromosome arms and a reduction in the amount of MEI-S332 at centromeres, while ATPsyn- α RNAi results in the lack of a MEI-S332 signal at centromeres (Figure 2F). We include zoomed insets to highlight these localisation patterns. These results suggest a specific function of ATPsyn- β like in chromosome arm cohesion. We propose a mechanism by which ATPsyn- α functions in the nuclear localisation of MEI-S332 to centromeres, whereas ATPsyn- β like restricts the loading of MEI-S332 to centromeres. We believe this role of ATPsyn- β like is different from the mechanism proposed for Bub1 (Boyarchuk et al 2007 JCB) as in addition to reduced MEI-S332 at centromeres, we observed MEI-S332 on chromosome arms, suggestion a function for ATPsyn- β like outside of the centromere. This requirement for MEI-S332 might be more comparable to the previously described function of the INCENP subunit of the Chromosome Passenger Complex (Resnick et al. 2006 Dev Cell). This data is presented in Figure 2 and has modified the Discussion as follows: *Remarkably, ATPsyn- β like specifies the enrichment of the cohesion protector MEI-S332 to centromeres at prometaphase I, perhaps comparable to the Chromosome Passenger Complex subunit INCENP²⁶. In contrast, ATPsyn- α appears to have a distinct function in the nuclear localisation of MEI-S332.*

Overall the mechanism of how this complex works with CENP-A is unclear, especially as the ORD/SUNN/SOLO complex is responsible for the same function later in prophase. This is all critical for one to assess the claims of the paper (e.g. the title of the paper): without further explanation, the data do not correlate a functional consequence of early prophase separation with fertility issues per se, especially if the separation problem reverts to a normal state prior to meiosis II.

Given that CENP-A was only reduced by 30%, we believe that many 16-cell cyst proceed through meiosis I and II and give rise mature sperm. This could explain why our quantitation of centromere foci at 'interphase' appeared normal. However, in line with segregation errors reported in Dunleavy et al. 2012, many cysts with reduced CENP-A arrest or fail in meiosis I or II. Overall this leads to a reduced rate of production of mature sperm and reduced fertility in males with depleted CENP-A in testes. To clarify these points, we have added the following to Results:

The number of centromere foci detected per nucleus at interphase did not differ from the control (3.7 compared to 3.6, $p=0.366$), possibly due to compensation by additional factors that maintain cohesion at this time or that the CENP-A depletion was less efficient in these cells.

And to the Discussion: We suggest that observed defects in cohesion lead to failed progression through meiosis I, and ultimately reduced in fertility or sterility.

5. There is a discrepancy between Figures 1C and 2D data points for S1/2a nuclei. The average is defined as 3.6 for control in 1C whereas it is 2.7 in 2D. Hence in 2D, an average of 3.4 foci is considered to be "high" when it is in wild type range in Figure 1. Although I understand that there is variability in RNAi experiments, the control CENP-A foci range should be consistent. The authors do not address this point in the text but it calls into question quantification of the other data points. It might be better to pool all the independent control experiments, define an average for S1/2a and all other stages and then compare all the mutants to the same range.

The control in 1C is the isogenic line host strain the the VDRC RNAi KK library (line 60000), whereas the control in 2D is the isogenic line host strain the TRIP RNAi library (line 36303). As suggested by the Reviewer, we have now pooled this data for independent control experiments and have defined an average value at S1/2a (3.072 foci, $n=222$ nuclei). Using this average value as the control, we report a significant difference in S1/S2a foci in CENP-A RNAi samples (Figure 1C) and we report a significant difference in S1/S2a foci in ATPsyn- α , ATPsyn- β and ATPsyn- γ RNAi samples but not ATPsyn- β like samples (Figure 1D).

6. Quantifications: There is a lack of the N used in experiments, and would be helpful in either figure legends or the main text, to list this alongside the P values. Some examples: Figure 1F needs an N. Figure 2E needs clarification of whether 20 testes were used in 1 experiment or 2 independent experiments. To be convincing, Fig. S2A requires numbers in terms of number of cells observed and whether the KD was observed in every cell or in a certain percentage, especially since the rest of the findings are hinged upon these KDs being consistent.

We have added the n number for each experiment to the Figure Legends and p values for each statistical analysis are either included in the text or clearly marked in Figures. To clarify, in 1E, counts for at least 12 testes were pooled from 2 independent RNAi experiments (crosses). For S2A, we include a quantitation of the efficiency of ATPsyn- α (90%), - β (100%), - β like (100%), - γ (100%) knockdown in S6 cells by IF staining with respective antibodies (n=2) (Figure S2B). In addition, we have added n numbers and standard deviations to Figure 3E.

7. Regarding the issues with centromeric localization of ATPsyn- β like (point #1, above), Fig. S4 D can be shown as single channel and with insets zoomed onto the centromeres. Again, an N is required for this panel to be convincing and maybe more pictures to support the claim. Figure 4D inset, should also be present in the single channels to be more convincing since the bright CENP-A staining makes it harder to see the faint GFP- ATPsyn- β like.

We now show centromeric localisation of both ATPsyn- α (with YFP-CENP-C) and GFP-ATPsyn- β like (with CENP-A) in Figure 4. We show multiple S6 nuclei, use zoomed insets to highlight overlapping foci and have calculated the Pearson Coefficient of colocalisation in each case (Figure S4C).

Minor concerns:

8. Fig S1E is referred to as S1F (line 119).

We have corrected this.

9. Are the authors confident that the ATPsyn- β like antibody does not recognize the ATPsyn-B? Some comment in the paper on this possibility should be made.

We believe this is unlikely as anti-ATPsyn- β like antibody was raised against two peptides unique to the N and C terminal extensions of ATPsyn- β like. The location of both peptides is now annotated in Figure S1D and is referred to in the methods section.

10. Authors make a strong claim that the N-terminus truncation has a very significant dominant negative effect and while it definitely has an effect, the numbers themselves do not look convincing especially if this domain is correlated with ATPsyn- β like function. Perhaps soften this claim, or test this notion further to strengthen it. For instance, test if this truncation can rescue CENP-A RNAi sister centromere cohesion defect while also measuring centromeric ATP-synB like levels.

We have softened this claim to reflect the subtlety of observed defects:

*Truncated GFP-CENP-A- Δ 118 localised to centromeres, but showed a dominant negative effect (*** $p < 0.0001$) on the number of centromeric foci at S5/6 compared to nuclei expressing full length GFP-CENP-A (Figure 5D). These results suggests that perturbation of the CENP-A N terminus can disrupt sister centromere cohesion in meiotic prophase I.*

11. Table S1 is not explained.

We have added a description of Table S1 to Supp Figure Legends.

12. Line 259: ATPsyn- β like localization was not shown in mitochondria?

Using a protocol that preserves the cytoplasm and staining with the anti-ATPsyn- β like antibody, we confirmed ATPsyn- β like localisation to mitochondria (shown in Figure S2B). In addition GFP-tagged ATPsyn- β like localises to mitochondria in S2 cultured cells (shown in Figure S4A).

13. Authors show that ATP-synB is expressed in testes Fig S1D, but in line 102, it says that this protein is repressed in testes. Is this stage specific repression or only germ cell specific?

Our qRT-PCR results (Figure S1G) and IF localisation studies (Figure S1E) indicate that ATPsyn- β is expressed in testes. A previous study from Wen et al. 2015 Mol Cell indicated that ATPsyn- β expression in testes is repressed 7 fold relative to whole adult flies. This study did not comment on the expression of ATPsyn- β -like. Therefore, we propose that ATPsyn- β -like might function in place of ATPsyn- β in testes. We have clarified this point in the Discussion as follows: *Given that canonical ATPsyn- β expression in testis is reduced compared to whole adults, ATPsyn- β -like might normally compensate for ATPsyn- β function.*

14. Fig 5A shows that ATPsyn- α has the strongest interaction with CENP-A even though in vivo staining does not show any nuclear ATP-syn α . Can the authors detect ATPsyn- α in germ cell nuclei at all? Is it reduced at centromeres in S5/6 in CENP-A and ATPsyn- α RNAi males?

Using a commercially available antibody, we can now localise ATPsyn- α to centromeres (costaining with YFP-CENP-C or anti-CENP-C, presented in Figure 4). Furthermore, we find that ATPsyn- α localisation at centromeres is reduced upon CENP-A RNAi in spermatocytes. This data is presented in Figure 4.

15. To be “nomenclature inclusive”, consider also providing the Drosophila world-centric names of the genes in parentheses in Table S2.

We have added the Drosophila name for ATPsyn- α , *bellwether*, to Table S2.

Reviewer #2 (Remarks to the Author):

The authors propose a novel function for the beta subunit of the ATP synthase in meiosis. If reproducible, this is an exciting discovery and should be of interest to a wide audience. The authors describe this meiosis function most strongly for a variant of ATP synthase beta called ATP syn beta-like. The manuscript is thorough but there are some deficiencies which need to be addressed before the data are acceptable for publication. In particular, the function of betalike in meiosis, that is to bind together CENP-A and MEI-S332 and the co-localization of betalike with CENP-A are preliminary and need further investigation. This may not even be the correct function of betalike, as outlined below, given the data the authors present.

We have modified our discussion of the major functions of ATPsyn- α and ATPsyn- β -like to reflect the notion that both are essential for ATP generation in mitochondrial, yet we have uncovered an additional nuclear function specific to centromeres.

Discussion: Here, in addition to an expected function in ATP synthesis, we report a function for ATPsyn- α and ATPsyn- β -like in male meiosis and fertility.

Furthermore, given that ATPsyn- β -like is expressed at larval L3 and pupal stages (Flybase RNA-Seq ModEncode), we are in agreement with the reviewer that ATPsyn- β -like likely has additional functions in development not addressed in this study.

Discussion: Moreover, given that ATPsyn- β -like is expressed at larval and pupal stages (modENCODE RNA-Seq), it is possible that ATPsyn- β -like adopts additional functions in development, which we have not address in this study.

Discussion: Intriguingly, depletion of ATP synthase F_1 subunits also disrupts sister chromatid arm cohesion and 4th homolog pairing/cohesion, suggesting additional global functions outside of the centromere.

1. Figure 2E and suppl data S2B: Although the drugs 2,4-DNP and oligomycin prevent ATP synthesis in the intact mitochondria, it would be important to rule out the role of ATP hydrolysis (of the isolated F_1 enzyme) in meiosis. This can be shown with other drugs that block ATP hydrolysis by the beta subunit-containing ATPase, even in the absence of the proton pumping membrane components of the ATP synthase.

To test if defects in ATP hydrolysis might result in the loss of centromere cohesion, we treated testes ex vivo with sodium azide, previously shown to block ATP hydrolysis of the isolated F_1 enzyme (Bowler et al. 2006 PNAS). Treatment of testes with 5 mM NaN_3 (concentration based on Sadiq et al. 2000 Mut Res) reduced ATP level to 26% (n=3) but no significant increase in centromere foci at S6 was observed ($p=0.2464$, 150 nuclei pooled from n=3). The data is presented in Figure S2C.

2. Figure 4A: The diffuse staining in Figure 4A of MEI-S332 does not demonstrate a lack of binding between MEI-S332 and CENP-A, it just shows a more diffuse localization of MEI-S332, but MEI-S332 is still present at the sites where CENP-A is localized (in the figure shown). To show there is a lack of binding between the two proteins in the absence of betalike, a co-IP needs to be performed.

To understand better effects of ATPsyn- β like on the recruitment and localisation of MEI-S332 to centromeres, we investigated in parallel MEI-S332 localisation in ATPsyn- α RNAi. We find that ATPsyn- α and ATPsyn- β like have different effects on MEI-S332 enrichment at centromeres. Knockdown of ATPsyn- β like results in the mislocalisation of MEI-S332 to chromosome arms and a reduction in the amount of MEI-S332 at centromeres, while ATPsyn- α RNAi results in the lack of a MEI-S332 signal at centromeres (Figure 2F). We include zoomed insets to highlight these localisation patterns. These results suggest a specific function of ATPsyn- β like in chromosome arm cohesion. We propose a mechanism by which ATPsyn- α functions in the nuclear localisation of MEI-S332 to centromeres, whereas ATPsyn- β like restricts the loading of MEI-S332 to centromeres. We believe this role of ATPsyn- β like is different from the mechanism proposed for Bub1 (Boyarchuk et al 2007 JCB) as in addition to reduced MEI-S332 at centromeres, we observed MEI-S332 on chromosome arms, suggestion a function for ATPsyn- β like outside of the centromere. This requirement for MEI-S332 might be more comparable to the previously described function of the INCENP subunit of the Chromosome Passenger Complex (Resnick et al. 2006 Dev Cell). This data is presented in Figure 2 and has modified the Discussion as follows: *Remarkably, ATPsyn- β like specifies the enrichment of the cohesion protector MEI-S332 to centromeres at prometaphase I, perhaps comparable to the Chromosome Passenger Complex subunit INCENP²⁶. In contrast, ATPsyn- α appears to have a distinct function in the nuclear localisation of MEI-S332.*

3. Figure 4B: The background subtraction images need to be shown and the details of the colocalization need to be better illustrated in the figure. Showing the diffuse staining for betalike is misleading and not parallel to the bar graph in 4C.

To support the localisation of ATPsyn- β like to centromeres, we focus on prophase I spermatocytes expressing GFP-ATPsyn- β like that have been fixed and costained with an anti-CENP-A antibody (labelled in the far red channel) (Figure 4C). We include a zoom inset of each separate channel as well as the merged channels at high resolution which shows the partial overlap of red and green signals. We have calculated an average Pearson colocalisation co-efficient for GFP-ATPsyn- β like and CENP-A foci at prophase I S6 stage of $0.5473 \pm \text{sd } 0.0896$ per nucleus ($n=20$ nuclei) (Figure S4C). A value of >0.5 indicates colocalisation. We believe that values are very close to 0.5 as GFP-ATPsyn- β like and CENP-A do not perfectly colocalise and the signals only partially overlap (foci are green, red and yellow).

In addition we present new data showing the colocalisation of ATPsyn- α (using an anti-ATPsyn- α antibody) and YFP-CENP-C (Figure 4B) or anti-CENP-C (Figure 4D) at centromeres. We have calculated an average Pearson colocalisation co-efficient for ATPsyn- α and YFP-CENP-C foci at prophase I S6 stage of $0.6443 \pm \text{sd } 0.1030$ per nucleus ($n=20$ nuclei) (Figure S4C). We believe that values are close to 0.5 as ATPsyn- α and CENP-C do not perfectly colocalise and the signals only partially overlap (foci are green, red and yellow).

4. 4D: Same as for 4C, the background subtracted images need to be shown. It is also not clear comparing figures 4B and D why the measurement of fluorescent intensity of betalike co-localizing with CENPA is used rather than the number or foci. The type of measurement used for the other figures (number of foci) should also be shown. What about using a nuclear localization signal to eliminate the cytosolic localization of betalike and show that it is reduced in the CENP-A RNAi?

We have added two new data sets showing colocalisation of either ATPsyn- α or GFP-ATPsyn β -like at centromeres (Figure 4B,4C). In CENP-A RNAi at 29°C ATPsyn- α ($n=46/50$ nuclei) and GFP-ATPsyn β -like ($n=47/47$) localisation is lost (pooled from 3 experiments). In addition, we confirmed that in line with Figure S1, the number of centromere foci per nucleus is increased these experiments (8.99 foci, 8.4 foci respectively) ($n=3$, $n=100$ nuclei counted from one experiment).

5. Figure 5A: It is disturbing that the binding between betalike and CENP-A is not direct, particularly since it is betalike reduction that really stops meiosis at early stages in figure 1E, much more than alpha. Therefore the mechanism of binding of MEI-S332 and CENP-A to each other depending on betalike does not make sense. To show the proposed role for betalike better, a mutation should be made in betalike that specifically prevents its co-IP with alpha (since that is the proposed mechanism the authors prefer). This mutant should disrupt binding of MEI-S332 and CENP-A (and co-localization at the centromere) and should halt meiosis at stage I as in Figure 1E. This still leaves the question open, however of why RNAi of alpha does not prevent the early stages of meiosis since by the authors reckoning, it is downstream of betalike in producing the binding between MEI-S332 and CENP-A. Could there be another binding partner or function of betalike that is more important for early meiosis?

We confirm knockdown of ATPsyn- α , - β , - β like, - γ and CENP-A (cid) at 25°C using qRT-PCR (Figure S1G). We note an almost 4 fold reduction in ATPsyn- α transcript compared to an approximate 10 fold reduction in ATPsyn- β like transcript, which might explain differences in the severity of phenotypes observed. We noted that while testes depleted for ATPsyn- α produce mature sperm, a number of abnormal cysts were observed at the MI and MII stages. To address this concern, we have performed the knockdown and cell cycle analysis of ATPsyn- α and - β like at 29°C (Figure 1E) and we show a representative image of perturbed 16-cell cysts (Figure S2B). Spermatocytes lacking ATPsyn- β like arrest at prometaphase I, whereas many spermatocytes lacking ATPsyn- α have perturbed nuclear morphology at prometaphase I yet proceed through meiosis I resulting in abnormal segregation and can produce mature sperm.

To understand better effects of ATPsyn- β like on the recruitment and localisation of MEI-S332 to centromeres, we investigated in parallel MEI-S332 localisation in ATPsyn- α RNAi. Knockdown of ATPsyn- α results in the lack of MEIS332 nuclear signal while knockdown of ATPsyn- β like results in the mislocalisation of MEIS332 to chromosome arms and a reduction in the amount of MEIS332 at centromeres. These results suggest a specific function of ATPsyn- β like in chromosome arm cohesion. We have modified the Discussion as follows: *Remarkably, ATPsyn- β like specifies the enrichment of the cohesion protector MEI-S322 to centromeres at prometaphase I, perhaps comparable to the Chromosome Passenger Complex subunit INCENP²⁶. In contrast, ATPsyn- α appears to have a distinct function in the nuclear localisation of MEI-S332.*

Reviewer #3 (Remarks to the Author):

Overall the paper contains some novel findings, and it is very interesting to see a role for ATP synthase subunits in sister centromere cohesion in meiosis, and a striking meiotic phenotype for depletion of the testis-specific ATPsynbeta isoform. Unfortunately there is insufficient detail in many places in the manuscript (methods, numbers for quantitative experiments, magnification/resolution for microscopy) for the reader to be sure that all the findings are robust. The detailed notes below are in order within the manuscript, not in order of priority.

line 80 states 1-3 cenp-A spots visible. line 84 states 3.4 dots in control. Which is correct?

The control in 1C is the isogenic line host strain the the VDRC RNAi KK library (line 60000), whereas the control in 2D is the isogenic line host strain the TRIP RNAi library (line 36303). As suggested by Reviewer 1, we have now pooled this data for independent control experiments and have defined an average value at S1/2a (3.072 foci, n=222 nuclei).

line 88, "problems with sister centromere cohesion are later resolved". What plausible mechanism is there for this? It would be very interesting to follow up on the analysis of numbers of foci and loss of cohesion with a genetic analysis of chromosome segregation defects (for conditions that give some fertility). At minimum you would expect to see aneuploidy in spermatids for those that progress through meiosis - this is easily scored by phase contrast analysis of nuclear size in early spermatids.

We that we achieved a knockdown of only 30%, we believe that many 16-cell cyst proceed through meiosis I and II and give rise mature sperm. This might be the ones we analysed as normal in our quantitation of centromere foci at 'interphase'. However, we also believe that many cysts arrest or fail in

meiosis I or II. Overall this leads to a reduced rate of production of mature sperm and reduced fertility. To address this concern, we have added the following to the Results:

The number of centromere foci detected per nucleus at interphase did not differ from the control (3.7 compared to 3.6, $p=0.366$), possibly due to compensation by additional factors that maintain cohesion at this time or that the CENP-A depletion was less efficient in these cells.

And to the Discussion:

We suggest that observed defects in cohesion lead to failed progression through meiosis I, and ultimately reduced fertility or sterility.

line 135, need to state in the text how the depletion was validated. RNAi depletion was validated by qRT-PCR (Figure S1G) and by IF (Figure S1H, S2B).

Figure 1 E, error bars should be SD, not SEM, n (of both nuclei and number of testes) should be stated somewhere.

In 1E, error bars on graphs show SEM, but we clarified in the legend the n number for testes (at least 12) and experiments (data pooled from 2 independent RNAi experiments i.e. crosses).

The data for Fig 1C should be shown in the same way as that for Figure 2B (in which case error bars are not needed at all).

This data is now presented in dot plots format showing all data points (Figure 2C).

Figure 1D, can you indicate which of the co-purifying bands are which protein.

As we performed a total analysis of pulldown fraction (rather than band extraction) we cannot precisely assign identities to the silver stained bands. The molecular weight of ATPsyn- α is 61kDa and ATPsyn- β is 57kDa and a number of bands of this size are visible on the gel.

Figure 1F clearly shows abnormal testes for the RNAi, but a phase contrast image as well is needed to show the arrest at prometaphase 1.

To better illustrate the observed arrest at prometaphase I, we have included a DAPI-stained image of whole testes depleted for ATPsyn β -like, as well as perturbed 16 cell cysts observed in ATPsyn- α depleted testes (Figure S2B).

Figure S1C, how can you have a cross-reacting band in the RT-PCR? What is it?

We have re-labelled this figure to read 'non-specific band'. We believe this band is a contaminating band amplified as a result of the nested PCR cycles.

Figure S4D, need to see unmerged images at higher zoom to convince me of the overlap between CENP-A and ATPsynbetalike. As presented there are so many ATPsynbetalike dots that it is almost inevitable that some will overlap the CENP-A dots.

To support the localisation of ATPsyn- β -like to centromeres, we focus on prophase I spermatocytes expressing GFP-ATPsyn- β -like that have been costained with an anti-CENP-A antibody (labelled in the far red channel) (Figure 4C). We include a zoom inset of each separate channel as well as the merged channels at high resolution which shows the partial overlap of red and green signals. We have calculated an average Pearson colocalisation co-efficient for GFP-ATPsyn- β -like and CENP-A foci at prophase I S6 stage of $0.5473 \pm \text{sd } 0.0896$ per nucleus ($n=20$ nuclei) (Figure S4C). A value of >0.5 indicates colocalisation. We believe that values are very close to 0.5 as GFP-ATPsyn- β -like and CENP-A do not perfectly colocalise and the signals only partially overlap.

In addition we present new data showing the colocalisation of ATPsyn- α (using an anti-ATPsyn- α antibody) and YFP-CENP-C (Figure 4B) or anti-CENP-C (Figure 4D) at centromeres. We have calculated an average Pearson colocalisation co-efficient for ATPsyn- α and YFP-CENP-C foci at prophase I S6 stage of $0.6443 \pm \text{sd } 0.1030$ per nucleus ($n=20$ nuclei) (Figure S4C). We believe that values are close to 0.5 as ATPsyn- α and CENP-C do not perfectly colocalise and the signals only partially overlap.

In addition, to further support the ATPsyn- α and ATPsyn- β -like localisation studies we show costaining of S6 nuclei with ATPsyn- α and ATPsyn- β -like antibodies and highlight overlapping discrete nuclear foci (Figure S4D).

Line 217 "Importantly, GFP-ATPsyn-βlike re-introduction into flies lacking one copy of ATPsyn-βlike partially restores sister centromere cohesion at S5/6 (Figure S4C). But you have not mentioned at all in the main text the evidence that ATPsyn-βlike heterozygotes have any defects to be rescued. The figure legend contains insufficient information regarding the actual numbers of foci per nucleus. Given the very subtle effect here more detail is needed. Is there a significant difference (as expected) between plus and minus rescue construct (only the differences to control are given p-values).

We have reviewed this data. We now compare 'ATPsyn-βlike^{+/-} heterozygote' to 'ATPsyn-βlike^{+/-} + GFP-ATPsyn-βlike rescue'. Although the mean number of centromere foci at S5/6 is decreased (6.68 foci to 6.43 foci, now indicated on the graph), no significant difference was calculated ($p=0.084$). We have modified the description of this experiment in the text as follows:

Re-introduction of GFP-tagged ATPsyn-βlike into flies lacking one copy of ATPsyn-βlike (ATPsyn-βlike^{+/-} heterozygote) that display a defect in sister centromere cohesion, reduced the mean number of centromere foci at S5/6 (Figure S4A). However, this reduction was not significant ($p=0.084$), indicating a partial functional rescue by the transgene.

In addition, we have added information on the ATPsyn-βlike P element insertion line to the methods.

Figure 4B. The quantitation of centromeric ATPsyn-βlike is not at all compelling. The entire testis is full of dots. Is this gene really expressed in all cells in testes (including the hub?), most testis-specifically expressed paralogues of ubiquitously expressed genes are expressed exclusively in the germline. RNA in situ hybridization would reveal which cells the gene is expressed in. Higher resolution microscopy is needed to convince me of the co-localisation.

See above (new data with quantitation now presented in Figure 4).

Figure 5A, there is an error in the labeling of this figure. I assume the first panel is pull down GST rather than pull down anti-his.

We have changed this label on the gel.

Line 250. Avoid using the phrase "more significant". More dramatic maybe, but give us the actual numbers of centromere foci, and the number of nuclei assayed, because the effect from the wild-type tagged construct is subtle. Can the N-terminal deletion cenpA line rescue the lethality of the null allele?

Given the subtle effects observed, we have softened our discussion of this data:

*Truncated GFP-CENP-A-Δ118 localised to centromeres, but showed a dominant negative effect ($***p<0.0001$) on the number of centromeric foci at S5/6 compared to nuclei expressing full length GFP-CENP-A (Figure 5D). These results suggests that perturbation of the CENP-A N terminus can disrupt sister centromere cohesion in meiotic prophase I.*

methods. Do not contain sufficient information that the experiments could be repeated.

We provide additional data in the methods section (addressed in points below).

- **more detail needed for the transgenic constructs. What genomic region was used for each - ie how much flanking dna etc. Were the tags at the c- or n-terminus. what vector was used for the transgenesis - p or site specific recombination?**
- Transgenic lines expressing N terminal tagged mCherry-ATPsyn-β, eGFP-ATPsyn-βlike or eGFP-Δ118-CENP-A under respective endogenous promoters were generated by transposable (P) element transformation of pCaSpeR5 vector in *w¹¹¹⁸* embryos (injection, selection and balancing by BestGene Inc). *ATPsyn-βlike* was amplified from wild type cDNA with 900 bp upstream of the start codon and 600 bp downstream of the stop codon. Δ118 *cid* (bp 354-678) was amplified from wild type cDNA with 413 bp upstream of the start codon and 417 bp downstream of the stop codon; a 3x glycine linker was placed between the GFP tag and the *cid* start codon.
- **the p-insertion line for atp-syn-beta-like should be referred to by its correct identifier, not Bloomington stock number.**
- We have modified this.

- **how long were flies left to lay eggs in the fertility test. Were all flies age-matched.**
- For fertility tests, two virgin age-matched males/females were crossed, allowed to lay eggs for two days and the number of adult progeny was scored after 20 days.
- **for protein prep from testes, what is in the hypotonic buffer?**
- The hypotonic buffer contained 10 mM HEPES, 1.5 mM NaCl, 1.5 mM MgCl₂, 0.1 mM EGTA, 1 mM DTT, 0.1 % Triton-X, 1 % protease inhibitor cocktail. Testes were incubated in this buffer for 10 minutes at 4 °C and the supernatant was collected at 1500 x g for 10 minutes at 4 °C.
- **What was GST conjugated to for the pull down?**
- GST was conjugated to glutathione agarose beads.
- **how exactly was the pull down done.**
- Protein extracts pooled from 500 adult testes were used per pull-down experiment; samples were diluted in 1 ml of interaction buffer (IB)(20 mM Tris-HCl pH 7.5, 300 mM NaCl, 0.5 mM EDTA, 0.05 % NP-40, 1 mM PMSF and 1 % protease inhibitor cocktail) and incubated for 2 hours at 4 °C and under 20 RPM with 10 µg of recombinant GST (control) or GST-CENP-A N-terminus (amino acids 1 – 126). 50 µl of glutathione agarose bead slurry was then added for a further 1 hour before the beads were collected by centrifugation at 700 x g and washed 3 x 15 minutes in 1 ml of wash buffer (IB containing 500 mM NaCl). Precipitated proteins from control and bait experiments were eluted by boiling (95 °C) in sample buffer.
- **What type of digest was done before MS (trypsin?).**
- Proteins were separated by SDS-PAGE then whole control (GST) and bait (GST-CENP-A N-terminus) gel lanes were excised and subjected to tryptic digestion.
- **How long was the fixation step for immunostaining?**
- Paraformaldehyde fixation was carried out for 10 minutes at room temperature. Methanol-Acetone fixation was carried out for 5 minutes in methanol followed by 2 minutes in acetone.
- **stats - t tests were used. Were any tests used to assess if the data are normally distributed?**
- We performed the D'Agostino & Pearson normality test in Prism. In most cases, the data was normally distributed ($P > 0.05$, passed the normality tests). However, in some experiments (e.g. Figure 2B) RNAi samples failed the normality test e.g. for ATPsyn-βlike-RNAi and ATPsyn-γ RNAi. We believe that this was due to presence of rare nuclei in which 15 or 16 centromere foci were counted i.e. outliers. Rather than remove this data from the analysis, we have kept these rare data points but have used of robust statistical t tests and large sample sizes to offset deviations from the ideal Gaussian distribution.
- **RT-PCR, much more detail needed - how many testes, how much RNA used for each assay. Was DNase treatment used? How many PCR cycles? PCR primer sequences needed. Control genes?**
- RNAs were isolated from 100 adult testes (< 2 day old) using RNeasy MiniElute kit (Qiagen) and DNase treated. For RT-PCR, 2 µg of RNA was reverse transcribed using SuperScript III Reverse Transcriptase kit (Invitrogen). 200 ng of cDNA was used in PCR reactions. Primer sets used for RT-PCR: ATPsyn-βlike: for atgttggtatcatgggctaaaatggct, rev: ttaatcttctttccgggtctttggcttg, ATPsyn-β: for: atgttcgcttacgtgctgca, rev: ctaggcagcttcccttgccagg. For qPCR, total testes cDNA was produced by reverse transcription of 600 ng of RNA in a 20 µl reaction using Applied Biosystems high capacity RNA-to-cDNA kit. DNA primers with 90-110 % efficiencies were selected: ATPsyn-α: for: ggcccttaacttgagcccg, rev: atggcaccggtacgcttgac; ATPsyn-β: for: ggtggctctcgatatgggc, rev: cggaagtactctgcaacgg; ATPsyn-βlike for: ggccatggattccaccgaagg, rev: gatgctcccaaacggcct; ATPsyn-γ: for: gatggtgtccgctgccaagt, rev: gccgatgccgtaaggacgag and cenp-a: for: gaagacggcaccgactacgg, rev: cgtcgaggaacgccgattgt. 10 ng of cDNA was used per reaction and qPCR was carried out using PowerUp SYBR Green MasterMix from Applied Biosystems. qPCR reactions were carried out according to the manufacturers fast cycling specifications on a StepOne Plus real time PCR system. Three technical replicates were carried out per experiment and data was pooled from two independent RNAi knockdowns. *gapdh* and *rp132* were used as internal control genes and fold change ($\Delta\Delta Ct$) was calculated using the comparative Ct method³⁰.
- **Scoring criteria for determining if two close together spots represent one or two foci, to reduce subjectivity. Is there a minimum size at which a dot counts as a focus?**
- We counted two foci if we could detect one 'back' pixel between two foci. We show an example in Figure 1B (zoom inset).

- **Was the counting done blind?**
- Regrettably, the counting was not performed blind. However, for many experiments, quantitations were counted independently by two different individuals (ED and CC) and results were highly reproducible between counts (e.g. Figure 2B).

typos and minor changes.

figure S1 legend, colons not semicolons in B.

Figure S2. These are whole spermatocytes, not just nuclei.

line 174, sites not site for the satellite location (on two chromosomes).

We have corrected these typos.

REVIEWERS' COMMENTS:

Reviewer #1 (Remarks to the Author):

In this revised version, Collins et al have improved and extended the study in a way that addressed satisfactorily most of the concerns that I had.

Some of the added experiments do bring up additional questions. It is not clear to me whether the authors know the answers to these questions and simply need to comment on them in the text, or rather that they should choose to perform additional experiments to answer them prior to publication. In any case, the authors would be well served to address them each in a satisfying way.

For instance, although the co-localisation of ATP-Synalpha and beta with CENP-A is greatly improved, there are variabilities within the cells that have not been addressed: Are the levels variable depending on cell stage? Is it a transient localization?

Another concern that arises from the new data is that the ATP-synalpha completely abrogates the staining of MEI-S332. This is striking, but confusing in that there is a more subtle phenotype in the alpha KD than the beta subunit which diffuses MEI-S332. It is a vital point considering that is essentially the only connection to mechanism in this paper.

Reviewer #2 (Remarks to the Author):

I have read carefully through the new manuscript and updated figures as well as the reviewer comments. I think the manuscript has been substantially improved, especially in the localization of CENP-A, ATP synthase alpha and ATP synthase beta-like. The data and discussion of mis-localization of MEI-S332 to centromeres have been expanded and clarified. The high resolution imaging improves the manuscript. Although the new functions attributable to ATP synthase F1 proteins are surprising, I now think that they are sufficiently robustly shown here to warrant publication.

There are some typos. I found two at least (in the second one, "than" is misspelled as "then"):

In testes

130 depleted for ATPsyn- α and - β , the number of spermatid cysts was not significantly
131 reduced ($p=0.147$ and $p=0.646$), however in testes depleted for ATPsyn- γ and the
132 number of spermatid cysts was significantly reduced ($****p<0.0001$) compared to the
133 control.

The disruption of arm cohesion was most

211 pronounced in ATPsyn- β -like-depleted nuclei in which greater than 5 foci per nucleus
212 was frequently observed.

Reviewer #3 (Remarks to the Author):

Overall the new experiments and details added have improved the paper, so that I now have much more confidence that the data supports the conclusions drawn. The methods section is dramatically improved. The imaging and co-localisation is now much more compelling. I have a few remaining very minor comments.

1) the fact that interphase cells in ATPsyn-alpha RNAi testes have normal sister cohesion despite the earlier defect is explained much better in the rebuttal letter than in the manuscript itself, and could be rephrased to be more clear. The assumption (based on the beta-like RNAi arrest) is that

cells with severe defects in prophase I don't progress to meiotic interphase. So the interphase cells seen must be derived from ones with normal cohesion in prophase I. I buy this. The alternative, that "SUNN/ORD/SOLO might function to maintain cohesion at this cell cycle stage" is possible for maintenance, but I can't see it being plausible for re-establishment of cohesion.

2) There is a new comment in the discussion relating to the fact that ATPsynbetalike is expressed in larvae and pupae - and suggesting other functions in development. While this is plausible (in the absence of further experiments), it is important to note that the expression pattern of beta-like is entirely consistent with expression and function exclusively in testes.

POINT BY POINT RESPONSE TO ISSUES RAISED BY REFEREES:

We thank all three reviewers for the careful consideration of our manuscript and helpful suggestions. We have made some minor edits to the manuscript to address any remaining queries.

Reviewer #1 (Remarks to the Author):

In this revised version, Collins et al have improved and extended the study in a way that addressed satisfactorily most of the concerns that I had. Some of the added experiments do bring up additional questions. It is not clear to me whether the authors know the answers to these questions and simply need to comment on them in the text, or rather that they should choose to perform additional experiments to answer them prior to publication. In any case, the authors would be well served to address them each in a satisfying way.

For instance, although the co-localisation of ATP-Syn α and beta with CENP-A is greatly improved, there are variabilities within the cells that have not been addressed: Are the levels variable depending on cell stage? Is it a transient localization?

We agree with the reviewer that there is some variability in ATPsyn- α and ATPsyn- β like signals at centromeres, both between individual nuclei and also between centromeres of the same nucleus. In general, we find that the strength of ATPsyn- α and ATPsyn- β like signals correlates with CENP-A/CENP-C intensities. We have focused on the prophase I S6 stage as at this time we can consistently detect ATPsyn- α and ATPsyn- β like at centromeres and we believe the images shown are representative. Of note, we could also detect ATPsyn- α and ATPsyn- β like at earlier stages of prophase I and we could no longer detect ATPsyn- α and ATPsyn- β like at centromeres after exit from from meiosis II (data not shown). To indicate such localisation dynamics, we have added the following sentence to the results p10/p11:

Additionally, we could detect ATPsyn- α and GFP-ATPsyn- β like at centromeres in early prophase I, however this localisation was lost after meiosis II (not shown).

Another concern that arises from the new data is that the ATP-syn α completely abrogates the staining of MEI-S332. This is striking, but confusing in that there is a more subtle phenotype in the alpha KD than the beta subunit which diffuses MEI-S332. It is a vital point considering that is essentially the only connection to mechanism in this paper.

The different effects of ATPsyn- α and ATPsyn- β like depletion on MEI-S332 localisation are indeed surprising. We suggest that this could be due to the fact that ATPsyn- β like RNAi cells arrest at prometaphase I resulting in a accumulation of mislocalised MEI-S332 on global chromatin. In contrast, many ATPsyn- α RNAi cells proceed through meiosis I with irregular nuclei and cell divisions and MEI-S332 does not accumulate on chromatin in the same manner. It is also possible that ATPsyn- β like has additional roles outside of the centromere, for example in global chromatin condensation that indirectly affect MEI-S332 localisation. A final possibility is that observed differences in MEI-S332 localisation may also be related to the differences in the efficiency of ATPsyn- α and ATPsyn- β like knockdowns. To expand on this idea, we have modified the Discussion on p13 as follows:

In contrast, ATPsyn- α appears to have a distinct function in the nuclear and centromeric localisation of MEI-S332. MEI-S332 mislocalisation to global chromatin in ATPsyn- β like depleted nuclei is particularly striking and might be a consequence of a sustained prometaphase I arrest or indicates a more general function of ATPsyn- β like on chromatin.

Reviewer #2 (Remarks to the Author):

I have read carefully through the new manuscript and updated figures as well as the reviewer comments. I think the manuscript has been substantially improved, especially in the

localization of CENP-A, ATP synthase alpha and ATP synthase beta-like. The data and discussion of mis-localization of MEI-S332 to centromeres have been expanded and clarified. The high resolution imaging improves the manuscript. Although the new functions attributable to ATP synthase F1 proteins are surprising, I now think that they are sufficiently robustly shown here to warrant publication.

There are some typos. I found two at least (in the second one, "than" is misspelled as "then"):

In testes

130 depleted for ATPsyn- α and - β , the number of spermatid cysts was not significantly
131 reduced ($p=0.147$ and $p=0.646$), however in testes depleted for ATPsyn- γ and the
132 number of spermatid cysts was significantly reduced ($****p<0.0001$) compared to the
133 control.

The disruption of arm cohesion was most

211 pronounced in ATPsyn- β -like-depleted nuclei in which greater than 5 foci per nucleus
212 was frequently observed.

We have corrected these typos.

Reviewer #3 (Remarks to the Author):

Overall the new experiments and details added have improved the paper, so that I now have much more confidence that the data supports the conclusions drawn. The methods section is dramatically improved. The imaging and co-localisation is now much more compelling. I have a few remaining very minor comments.

1) the fact that interphase cells in ATPsyn-alpha RNAi testes have normal sister cohesion despite the earlier defect is explained much better in the rebuttal letter than in the manuscript itself, and could be rephrased to be more clear. The assumption (based on the beta-like RNAi arrest) is that cells with severe defects in prophase I don't progress to meiotic interphase. So the interphase cells seen must be derived from ones with normal cohesion in prophase I. I buy this. The alternative, that "SUNN/ORD/SOLO might function to maintain cohesion at this cell cycle stage" is possible for maintenance, but I can't see it being plausible for re-establishment of cohesion.

Here we presume that Reviewer 3 is referring to our observation (in Figure 1C) that sister centromere cohesion is normal in CENP-A RNAi at interphase (and not ATPsyn- α RNAi that have a significantly increased number of centromeric foci per nucleus at interphase ($P<0.0001$) as shown in Figure 2D). To clarify this point we have modified the Results on page 4/5 as follows:

It is possible that these interphase cells derived from prophase I cells with normal cohesion, or that these cells arise due to compensation by additional factors that maintain cohesion at this time. Additionally, it is also possible that the CENP-A depletion was less efficient in these cells.

2) There is a new comment in the discussion relating to the fact that ATPsynbetalike is expressed in larvae and pupae - and suggesting other functions in development. While this is plausible (in the absence of further experiments), it is important to note that the expression pattern of beta-like is entirely consistent with expression and function exclusively in testes.

To reflect this point, we have modified our Discussion page 12 as follows:

Moreover, although the expression pattern of ATPsyn- β like is entirely consistent with a testis-specific function, we note ATPsyn- β like expression at larval and pupal stages (modENCODE RNA-Seq). This raises the possibility that ATPsyn- β like adopts additional functions in development, which we have not addressed in this study.